# Heat waves monitoring over West African cities: uncertainties, characterization and recent trends

Cedric G. Ngoungue Langue[1,2], Christophe Lavaysse[2,3], Mathieu Vrac[4], and Cyrille Flamant[1]

[1]Laboratoire Atmosphères, Milieux, Observations Spatiales (LATMOS) - UMR 8190 CNRS/Sorbonne Université/UVSQ, 78280 Guyancourt, France.
[2]Université Grenoble Alpes, CNRS, IRD, G-INP, IGE,38000 Grenoble, France
[3]European Commission, Joint Research Centre (JRC), 21027 Ispra, VA, Italy
[4]Laboratoire des Sciences du Climat et de l'Environnement, CEA Saclay l'Orme des Merisiers, UMR 8212 CEA-CNRS-UVSQ, Université Paris-Saclay & IPSL, 91191 Gif-sur-Yvette, France.

**Correspondence:** Ngoungue Langue Cedric Gacial (cedric-gacial.ngoungue-langue@latmos.ipsl.fr)

**Abstract.**

Heat waves can be one of the most dangerous climatic hazards affecting the planet; having dramatic impacts on the health of humans and natural ecosystems as well as on anthropogenic activities, infrastructures and economy. Based on climatic conditions in West Africa, the urban centers of the region appear to be vulnerable to heat waves. The goals of this work is firstly to assess the potential uncertainties encountered in heat waves detection; and secondly analyse their recent trend in West Africa cities during the period 1993-2020. This is done using two state-of-the-art reanalysis products, namely ERA5 and MERRA, as well as two local station datasets, namely Yoff Dakar in Senegal and Aéroport Félix Houphouët Boigny Abidjan in Ivory Coast. An estimate of station data from reanalyses is processed using an interpolation technique : the nearest neighbor to the station with a land sea mask >=0.5; the interpolated temperatures from local station in Dakar and Abidjan, show slightly better correlation with ERA5 than MERRA. Three types of uncertainties are discussed: the first type of uncertainty is related to the reanalyses themselves, the second is related to the sensitivity of heat waves frequency and duration to the threshold values used to monitor them; and the last one is linked to the choice of indicators and the methodology used to define heat waves. Three sorts of heat waves have been analysed, namely those occurring during daytime, nighttime and both daytime and nighttime concomitantly. Four indicators have been used to analyse heat waves based on 2-m temperature, humidity, 10-m wind or a combination of these. We found that humidity plays an important role in nighttime events; concomitant events detected with wet-bulb temperature are more frequent and located over the north Sahel. Strong and more persistent heat waves are found in the CONT region. For all indicators, we identified 6 years with a significantly higher frequency of events (1998, 2005, 2010, 2016, 2019 and 2020) possibly due to higher sea surface temperatures in the equatorial Atlantic ocean corresponding to El Nino events for some years. A significant increase in the frequency, duration and intensity of heat waves in the cities has been observed during the last decade(2012-2020); this is thought to be a consequence of climate change acting on extreme events.

**Keywords :** heat extremes, El Niño, reanalysis data, climate change, climatic regions, West Africa.

# 1 Introduction

Since the industrial revolution, the Earth is experiencing a global warming related to human activity ((Hartmann et al., 2013); Intergovernmental Panel on Climate Change-IPCC-report 2021). The last report of IPCC shows that this warming will exceed $1.5°C$ with respect to the IPCC baseline 1850-1900 under different Shared Socio-economic Pathways (SSP) in 2100 if the rate of greenhouse gas emissions is not reduced. This warming climate contributes to the occurrence of extreme events, but also tends to reinforce their intensity ((Fischer and Schär, 2010; Engdaw et al., 2022); IPCC report, 2021). Heat waves appear as one of the most dangerous climatic hazards affecting the planet due to their impacts on several sectors (Perkins, 2015). The health sector is the most affected; heat waves act on the thermal comfort of the body leading to an increase in morbidity, respiratory and cardiovascular diseases among the most vulnerable population (children and eldery) (e.g., Huynen et al., 2001; Braga et al., 2002; Hajat et al., 2007; Kovats and Hajat, 2008; Anderson and Bell, 2009; Gasparrini and Armstrong, 2011; Rocklöv et al., 2014). Heat waves are "silent killers" because their impacts on human health are not usually instantaneous (Loughnan, 2014). In 2003, an intense heat wave occurred in France, killing more than 14000 people (Fouillet et al., 2006). During this event, temperatures sometimes reached 37°C, a record since 1950. This event was very persistent and lasted two weeks in France. In addition to this event, the Russian heat wave in 2010 caused numerous destruction (dysfunction of railway stations, interruption of energy production) and more than 11000 deaths (Shaposhnikov et al., 2014). Temperatures sometimes reached 38°C and generated huge fires in the neighboring regions of Moscow; and a high concentration of carbon monoxide in the troposphere. In April 2010, northern Africa was affected by a severe heat wave with daily maximum temperatures frequently exceeding 40°C and daily minimum temperatures over 27°C for more than five consecutive days (Largeron et al., 2020).

Heat waves are natural disasters often associated with an increase in daytime and/or nighttime temperatures. More generally, they are defined as a period of consecutive days for which the temperatures are much hotter than the normal. There is no universal formulation describing a heat wave, however a definition could be made according to the context of the study (health, environment, infrastructure, agriculture, energy supply). From a physiological point of view, the severity of a heat wave is measured through its duration and intensity.

West Africa experiences a very hot and dry climate over the Sahel region, and a hot and humid climate over the Guinea coast. The climatic conditions over West Africa make the region vulnerable to heat waves when it comes to the health of humans and natural ecosystems, but also agriculture. Many studies on heat waves have been carried out in Europe. However, heat waves in Africa are still not well documented. Moron et al. (2016) analyzed the trends of extreme temperatures in the northern tropical Africa from observations and reconstructed data. They show that heat waves indices over the region were highly correlated with the El Niño-southern Oscillation indices (ENSO) Barbier et al. (2018) investigated the intraseasonal variability of large-scale heat waves during the spring using the Berkeley Earth Surface Temperature (BEST) gridded dataset and reanalyses: the European Centre for Medium-Range Weather Forecasts interim reanalysis (ERA-Interim), Modern-Era Retrospective analysis for Research and Applications (MERRA, see section Data for more details) and the National Centers for Environmental Prediction reanalysis (NCEP-2). They defined heat waves using anomalies of minimum/maximum values of

the 2-m temperature. They found some discrepancies in the characteristics, variability and climatic trends of heat waves in the different products. Largeron et al. (2020) analysed the April 2010 heat wave in North Africa using both the BEST dataset and climate simulations from the atmospheric component of the Centre National de Recherches Météorologiques (CNRM) climate model. They showed a strong link between heat waves over the Sahara and the incoming heat surface fluxes. Another important result of this work, is the radiative effect of water vapor on minimum temperatures during the heat wave period. This can lead to extreme heat conditions during the night and cause death to eldery. Guigma et al. (2020) analysed the characteristics and thermodynamics of Sahelian heat waves using different thermal indices based on temperature, wind speed, relative humidity derived from ERA5 reanalysis (see section Data for more details). They found that most of the regions in the Sahel experience on average one or two heat waves per year with a duration of 3-5 days and of severe magnitude. They have also shown that the eastern Sahel experienced more frequent and longer events. They identified heat advection and greenhouse effect of moisture as the main drivers of Sahelian heat waves. Some of the previCedric1991 ous studies conducted over the Sahelian band, only use the daily maximum and minimum temperatures (e.g. , Moron et al., 2016; Barbier et al., 2018) for the detection of heat waves, thereby ignoring the potential influence of humidity and wind speed. Others take into account the effect of humidity in the heat wave definition (e.g. , Guigma et al., 2020), but information about the interannual and seasonal variability of events detected are missing; even though this is very important for policy makers and governments to take into account in order to develop early alert systems. Recently, Engdaw et al. (2022) studied the trends of heat waves over Africa during the period 1980-2018 using observations from the Climate Research Unit version 4.03 (CRU TS4.03) and BEST datasets as well as the following reanalysis datasets: ERA5, MERRA-2 and the Japanese Meteorological Agency's 55-year reanalysis (JRA-55). They highlighted large differences in both trend and temporal evolution of heat wave indices between the different reanalyses. They found a peak of heat over northern and western Africa in 2010 as well as in 2016 over eastern and southern Africa. They noticed a significant warming and an increase in heat wave occurrence in all the regions in Africa. However, Engdaw et al. (2022) focused only on dry heat waves over a large domain of west Africa [20°W-20°E,10°S-15°N]; the duration of heat waves were not addressed, nor the evolution of wet heat waves.

The most lethal heat waves are not only due to high temperatures, but also to the effect of humidity (Steadman, 1979a, b); hot and humid conditions (as is the case in coastal regions) can be more dangerous than equivalently hot but dry conditions (Wehner et al., 2017). Wet heat waves, which are the most dangerous for human health, were not investigated in the previous works. Following Steadman (1979a, b), one can legitimately wonder about the effect of humidity on the frequency of heat waves and on the evolution of humid heat waves in west African cities. Based on previous studies, many definitions of heat wave have been proposed, leading to different results. Indeed there is no universal definition of a heat wave; depending on the research applications, some indicators and definitions can be adopted. Thus, we can question the potential sources of uncertainty found in heat waves analysis.

The goals of this paper are: (i) to highlight the potential uncertainties encountered in the heat wave detection process, and (ii) to analyse the recent trend and characteristics of dry and wet heat waves over a selection of West African cities grouped in

climatic regions. To achieve these objectives, we first assess the biases in the reanalyses (ERA5, MERRA) using ERA5 as a reference; then, a sensitivity analysis of the frequency of heat waves with respect to the threshold values, indicators and methods applied to define heat waves is addressed. Finally, we assess the spatial and temporal variability (seasonal and interannual) of heat waves and their characteristics in different climatic regions over West Africa.

The remainder of this article is organised as follows: in section 2, we present the regions of interest and the data used for this work; the description of the methodology is also provided. Section 3 contains the main results of this study following the methodology described in section 2. In section 4, the uncertainty in the reanalyses and the origin of coastal heat waves are discussed. Section 5 provides a conclusion and some perspectives for future works.

## 2 Region of interest, Data and Methods

### 2.1 Region of interest

The current study is conducted over west Africa which is located over the domain [5-20°N, 15°W-10°E], and spans from the Atlantic coast to Chad, and from the Gulf of Guinea to the southern fringes of the Sahara desert [Fig1]. The climate in West Africa is mostly influenced by the West African monsoon flux which governs the rainy season and thus the rain-fed agriculture. The West Africa region has a semi-arid and hot climate with a dry season (Koppen classification BSh or BShw). This climate corresponds to an alternation between a short wet season and a very long dry season. The West Africa region shows high climate variability at regional- and local- scale. In this study, we are interested in the coastal and continental parts of West Africa. We have therefore identified three regions based on their location and climate variability on which we have conducted our analyses. The choice of these regions is coherent with Moron et al. (2016) who used a hierarchical clustering approach to define some city blocks over West Africa. The fifteen cities investigated here were classified into the three following regions :

- Continental zone (**CONT** hereafter) including the cities of Bamako, Ouagadougou and Niamey [Fig1];

- Coastal atlantic zone (**ATL** hereafter) including the cities of Dakar, Nouakchott, Monrovia and Conakry [Fig1];

- Coastal Guinean zone (**GU** hereafter) including the cities of Yamoussoukro, Abidjan, Lomé, Abuja, Lagos, Accra, Cotonou and Douala [Fig1].

The CONT and GU regions are very similar to the clusters found by Moron et al. (2016) (see figure under the title 'Clusters membership' in Moron et al. (2016) ). The ATL region is a specific case because not all cities belonging to the region are present in the clusters defined by Moron et al. (2016). Therefore, we analyzed the spatial variability of heat wave characteristics for each city. In this way, we found a consistent pattern across cities (see [Figreffig:S1] in the supplementary material for maximum T2m values using the $90^{th}$ percentile as a threshold); and we grouped them to form the ATL block.

## 2.2  Data

Reanalysis products are often taken as an alternative solution to observational weather and climate data due to availability and accessibility problems, particularly in data-sparse regions such as Africa (Gleixner et al., 2020). In this work, to access information with a regular spatial grid and a large horizontal coverage, we used two state-of-the-art reanalysis products: the fifth-generation European Center for Medium-Range Weather Forecasts (ECMWF) reanalysis (ERA5; (Hersbach et al., 2020)); and the Modern-Era Retrospective analysis for Research and Applications, version 2 (MERRA-2; (Gelaro et al., 2017)) from the National Oceanic Atmospheric Administration (NOAA); (in the following, we will use "MERRA" to refer to MERRA-2). ERA5 reanalysis has a native spatial resolution of 0.28125 degree ($\sim 31\ km$) with 137 hybrid sigma/pressure levels from the surface up to $80\ km$, yet downloaded data are interpolated to a regular latitude/longitude grid of 0.25° x 0.25°. ERA5 is produced using 4D-Var data assimilation and the Cycle 41r2 (Cy41r2) of the ECMWF Integrated Forecast System (IFS), which was operational in 2016. MERRA reanalysis has a spatial resolution of 0.625°x0.5° with 42 standard pressure levels. MERRA is using an upgraded version of the Goddard Earth Observing System Model, Version 5 (GEOS-5) data assimilation system and the Global Statistical Interpolation (GSI) analysis scheme of Wu et al. (2002). MERRA is produced using a 3D-Var data assimilation algorithm. These two reanalyses dataset are assessed through the Climserv database from the Institut Pierre Simon Laplace (IPSL) server. To be consistent in our analyses, we transformed the spatial resolution of MERRA from 0.625°x0.5° to 0.25°x0.25° to match the one of ERA5; this is done using a first order conservative interpolation. We use hourly data covering the period going from 1 January 1993 to 31 December 2020 both for ERA5 and MERRA. Our choice of ERA5 and MERRA to conduct this study is supported by some previous work showing that these two reanalyses are part of the most relevant used in Africa regions (e.g. , Barbier et al., 2018; Ngoungue Langue et al., 2021; Engdaw et al., 2022). As the main objective here is to process heat waves detection, we focus on atmospheric variables at the surface such as 2-meter temperature ($T2m$), 2-meter relative humidity (Rh), 2-meter dew-point temperature, 2-meter specific humidity, 10-meter wind components and water vapor pressure ($e$) from which wet bulb temperature ($Tw$) and Apparent Temperature ($AT$; (McGregor et al., 2015)) were derived. These atmospheric variables have a significant impact on human thermal comfort (McGregor et al., 2015). Daily minimum and maximum values were calculated for $T2m, Tw, AT$ and the Universal thermal Comfort Index ($UTCI$; (Di Napoli et al., 2021)). $AT$ is similar to the heat index developed by Steadman (1984). The climate variables $e, Tw, AT$ and $Rh$ were calculated using the following formulas:

$$\mathbf{e = 6.1121 * \exp(\frac{17.502 * T}{240.97 + T})} \tag{1}$$

(Buck, 1981; Alduchov and Eskridge, 1996)

$$\mathbf{Tw = T * atan[A(Rh + B)^{\frac{1}{2}}] + atan(T + Rh) - atan(Rh - C) + D * (Rh)^{\frac{3}{2}} * (atan(E * Rh)) - F} \tag{2}$$

(Stull, 2011), ( Rh is used in percentage, for example 32 for Rh=32%)

$$\mathbf{AT = T + 0.33 * e - 0.70 * W_s - 4.00} \tag{3}$$

(McGregor et al., 2015)

Rh is computed differently based on the variables available in the products. The first formula is used to compute Rh in ERA5, and the second is used for MERRA.

$$\mathbf{Rh = 100 * \frac{\exp(\dfrac{a * T_d}{b + T_d})}{\exp(\dfrac{a * T}{b + T})}} \tag{4}$$

(August, 1828; Magnus, 1844; Alduchov and Eskridge, 1996)

$$\mathbf{Rh = 0.263 * p * q * [\exp(\frac{17.67 * (T - T_0)}{T - 29.65})]^{-1}} \tag{5}$$

https://earthscience.stackexchange.com/questions/2360/how-do-i-convert-specific-humidity-to-relative-humidity

$\mathbf{a} = 17.625, \mathbf{b} = 243.04, \mathbf{A} = 0.151977, \mathbf{B} = 8.313659, \mathbf{C} = 1.676331, \mathbf{D} = 0.00391838, \mathbf{E} = 0.023101, \mathbf{F} = 4.686035, \mathbf{T_0} = 273.16K$

Where $\mathbf{T}$(°C), $\mathbf{T_d}$(°C), $\mathbf{T_0}$(K), $\mathbf{p}$(hPa), $\mathbf{W_s}$(m/s) and $\mathbf{q}$ are respectively the ambient temperature, dew-point temperature, reference temperature, pressure, wind speed and specific humidity.

The land sea mask dataset used in this work was derived from ERA5 reanalysis; it can be accessed on the Copernicus Data Store (CDS). T2m daily maximum and minimum observations at Dakar-Yoff station in Senegal and Aéroport Félix Houphouët Boigny (FHB) station in Ivory Coast were used to evaluate our interpolation method. This is due to the fact that we do not have access to other station data in these regions. The data from Dakar-Yoff extend from 1 January 1973 to 31 December 2018 and contain almost 16% of missing values; and the data from Aéroport FHB extend from 1 January 2005 to 31 December 2017 and contain 0.35% missing values. These data were provided by colleagues from the Agence Nationale de l'Aviation Civile et de la Météorologie (ANACIM) for the Dakar-Yoff station, and from the Institut des Géosciences de l'Environnement (IGE) for the Aéroport FHB station.

## 2.3  Methods

### 2.3.1  Estimation of atmospheric variables at the scale of cities

Reanalysis dataset used for weather studies are generally run at global scale, therefore information at local scale is missing in many regions; this is a critical issue in regions where there is a lack of observation stations as is the case of African cities. To overcome this problem, sometimes downscaling methods can be used. In this work, we study phenomena at the

scale of the cities and reanalyses (ERA5 and MERRA) have too coarse a spatial resolution. The scales of the reanalyses are more representative of the spatial variability of a heat wave occuring in a city than an isolated local station. Nevertheless, some validation of the test stations needs to be done, in particular to find the best interpolation technique to estimate local temperatures from the reanalyses. This is especially important over the coastal regions. Indeed, most of the cities used in this study are located along the coast and influenced by the ocean air masses (see [Fig1]). The evaluation of the spatial variability of the correlation between the local scale variable (station) and reanalyses (ERA5) for T2m for example, showed high correlation values over the continent [FigS2] (Dakar, Abidjan). This suggests that the station data are well correlated with ERA5 grid points which are located on the continent; it is therefore necessary to know whether an ERA5 grid point is over the continent or not before applying an interpolation technique. To estimate the proportion of land on a grid point, we used the land sea mask (lsm) with values ranging from 0 to 1. The land sea mask is a measure of the land occupation on a grid point. A lsm of 0 means no land (a grid point located in the ocean), and a lsm of 1 means that the model cell is fully covered by land. Therefore, to estimate the climate variables over the cities from reanalyses, we use the nearest grid point of reanalyses to the station which satisfies an lsm equal or greater than 0.5 (see [Table1] for lsm values of all the cities considered in this study). This approach was chosen after evaluating different methods such as (see [FigS3a] for more details) :

– a bilinear interpolation using the four nearest grid points of reanalyses around the station [FigS3a (a,d)];

– a linear gradient approach which considers that the gradient of temperature is constant between two grid points based on a linear interpolation with a condition on the lsm value (>=0.5) [FigS3a (c,f)];

– the selection of the nearest grid point of reanalyses from the station with different values of lsm (>=0.5, 0.75 and 1; we only show for lsm>=0.5) [FigS3a (b,e)].

– a dynamical interpolation approach taking into account the effect of winds (not shown).

The interpolation method was applied to ERA5 and MERRA, and the resulting estimated data were compared to the station data by correlation analysis. We found that ERA5 appears to be slightly better than MERRA at both stations (Dakar and Abidjan) for minimum and maximum $T2m$ values [FigS3b].

### 2.3.2 Heat wave detection

Heat waves are usually defined as consecutive days of extremely hot temperatures above a threshold temperature value (e.g., Tan et al., 2010; Gasparrini and Armstrong, 2011; Perkins and Alexander, 2013; Wang et al., 2019). Many factors can affect the definition of a heat wave, including the end-user sectors (human health, infrastructures, transport, agriculture) and also the climatic conditions of the regions (Perkins and Alexander, 2013). Therefore, there is no universal and standard definition of a heat wave (Perkins, 2015; Oueslati et al., 2017; Shafiei Shiva et al., 2019). Different thresholds, duration and indicators contribute to the divergence in the definition of heat waves (Smith et al., 2013). Heat waves can be defined from daily meteorological variables such as daily raw temperature ($T_{min}, T_{mean}$ and $T_{max}$) (e.g., Fontaine et al., 2013; Beniston et al., 2017; Ceccherini et al., 2017; Déqué et al., 2017; Batté et al., 2018; Barbier et al., 2018; Lavaysse et al., 2018; Engdaw et al., 2022), mean daily

wet bulb temperature (Yu et al., 2021) or heat stress indices (e.g., Robinson, 2001; Fischer and Schär, 2010; Perkins et al., 2012a; Guigma et al., 2020) using relative or absolute thresholds. The use of absolute thresholds is well suited to detect heat

waves during the year in regions where the seasonal cycle is well marked. In mid-latitudes for example, the seasonal thermal amplitude of $T2m$ is large, approximately 20°C. In tropical regions this method is not suitable since the seasonal thermal amplitude is strongly reduced (6°C). Therefore, a relative threshold for heat waves detection is adopted in our study since our region of interest is West Africa. Some authors use the daily anomalies of temperature to define heat waves (e.g., Stefanon et al., 2012; Barbier et al., 2018). Most of the previous studies are focused on daytime or nighttime heat waves, ignoring events

which occur during the day and night concomitantly. These type of heat waves are very dangerous for human health because the body suffers from heat stress during the day and night (Lavaysse et al., 2018). In our case, we defined 3 methods to detect specific types of heat waves (namely those occurring during daytime, nighttime and both daytime and nighttime concomitantly) using the daily minimum and maximum values of: $T2m$ ($T2m_{min}, T2m_{max}$), $Tw$ ($Tw_{min}, Tw_{max}$), $AT$ ($AT_{min}, AT_{max}$) and $UTCI$ ($UTCI_{min}, UTCI_{max}$) as indicators. The selected atmospheric variables have been used for heat wave detection

in previous studies; they take in account some key parameters (air temperature, wind, humidity, radiant temperature) to assess the body heat stress and they are easy to compute. The methods applied are defined below :

- **Method 1:** A heat wave is defined as a consecutive period of at least 3 days during which the daily maximum value of an indicator exceeds the calendar $90^{th}$ percentile of daily maximum values of the indicator computed over the whole period (see **HW1** in [Fig2]). This approach is useful for monitoring daytime heat wave events. Daytime events will be
more associated with incoming solar radiation;

- **Method 2:** A heat wave is defined as a consecutive period of at least 3 days during which the daily minimum value of an indicator exceeds the calendar $90^{th}$ percentile of daily mininimum values of the indicator computed over the whole period (see **HW2** in [Fig2]). This approach is useful for monitoring nighttime heat wave events. Nighttime events can be related to the moisture content of the region;

- **Method 3:** A heat wave is defined as a consecutive period of at least 3 days during which daily minimum and maximum values of an indicator exceed the calendar $90^{th}$ percentiles of daily minimum and maximum values respectively (see **HW3** in [Fig2]). This method is most appropriate for extreme events that occur both during the day and at night, and are very harmful for human health.

The $90^{th}$ percentile is calculated for each calendar day of the year using an 11-day moving window centered on the day

under study. The use of a moving window allows the seasonal cycle to be taken into account in the calculation of percentiles. The use of a relative threshold is more appropriate as it is easily replicable in other regions. When two heat wave events are separated by one day with an indicator value below the daily $90^{th}$ percentile, they are pooled together to form a single event (see [Fig2]).

### 2.3.3 Heat wave characteristics

Once a heat wave is detected, some key characteristics are derived, namely duration and intensity. Some studies use the Heat Wave Magnitude Index daily (HWMId) to assess the severity of heat waves (e.g., Russo et al., 2016; Ceccherini et al., 2017). The HWMId focuses on strong heat waves; using this measure, the total intensity of all detected events cannot be accessed. In our study, the methodology applied to compute the duration and intensity of heat waves has been developed by Lavaysse et al. (2018) for monitoring extreme temperatures in Europe. We define the heat wave duration as the total number of hot days in heat waves. Hot days are heat wave days with daily values of the indicators above the daily thresholds. The heat wave duration is computed using the following expression:

$$\mathbf{duration} = \sum_{i=1}^{N} \sum_{j}^{d} \delta_j \tag{6}$$

where $\delta_j = 1$ if $T_j > $ daily $90^{th}$ percentile and $\delta_j = 0$ if $T_j < $ daily $90^{th}$ percentile, $N$ represents the total number of heat waves per grid point and d the number of hot days in a heat wave. The kronecker $\delta_j$ is used here because we pooled heat waves separated by 1 day together to form single events. For example, two heat waves of 4 and 3 days respectively separated by 1 day below the threshold will be counted as a single event with a duration of 7 days. For each block defined previously, the duration of heat waves is computed as the average of the duration of heat waves over the cities belonging to the same block. This also applies for the intensity.

The intensity of a heat wave has been defined as the sum of the daily exceedances of daily values of indicators to the climatological daily threshold in a sequence of hot days. This study is part of the project Agence National de la Recherche STEWARd (STatistical Early WArning systems of weather-related Risks from probabilistic forecasts, over cities in West Africa) project which focuses on the human impacts of climate extremes. Therefore, the climatological daily threshold is chosen to be constant over the whole period; and it is defined as the minimum of the daily climatology thresholds over the study period. This approach allows us to properly assess the severity of a heat wave and its potential human impacts. The expression of the intensity is given by :

$$\mathbf{I_1} = \sum_{t=1}^{T} \mathbf{bool_{max,t,w}} * (\mathbf{X_{max,t,w}} - \min(\mathbf{Q_{max,w}})) \tag{7}$$

$$\mathbf{I_2} = \sum_{t=1}^{T} \mathbf{bool_{min,t,w}} * (\mathbf{X_{min,t,w}} - \min(\mathbf{Q_{min,w}})) \tag{8}$$

$$\mathbf{I_3} = \sum_{t=1}^{T} \mathbf{bool_{min-max,t,w}} * (\mathbf{X_{max,t,w}} - \min(\mathbf{Q_{max,w}})) + \sum_{t=1}^{T} \mathbf{bool_{min-max,t,w}} * (\mathbf{X_{min,t,w}} - \min(\mathbf{Q_{min,w}})) \tag{9}$$

$I_1, I_2, I_3$ are intensities associated to $HW_1, HW_2, HW_3$ respectively. $X_{max,t,w}/X_{min,t,w}$ are daily maximum/minimum values of indicators at the grid point w. $Q_{max,w}/Q_{min,w}$ represents daily maximum/minimum threshold of the indicators at the grid point w. $bool_{max,t,w}$ and $bool_{min,t,w}$ are boolean time series which contain 0 if the day is not part of a heat wave, and 1 if the day is part of a heat wave for maximum and minimum daily values of the indicators respectively. $bool_{min-max,t,w}$ is a boolean time series which indicates 1 if the day is part of a heat wave detected simultaneously with both the minimum and maximum values of indicators, and 0 if it is not the case. T is the length in days of the study period. The mean duration and intensity are used to assess the severity of the heat wave.

### 2.3.4  Evaluation of the products using statistical metrics (Hit rate, ACC, GSS)

Most regions in Africa suffer from a lack of observations due to a small number of available weather stations. To access information over a large domain, we use ERA5 and MERRA reanalysis datasets which are very consistent in representing large-scale processes in the Saharan area (Ngoungue Langue et al., 2021). The coherence of reanalyses at regional scale was assessed using statistical metrics such as the Hit rate, the Anomaly Correlation Coefficient (ACC) and the Gilbert skill score (GSS). The Hit rate and GSS are used to evaluate hot days in the reanalyses.

#### – Hit rate

The Hit rate, also known as the "hit", is a measure of the fraction of events detected in an evaluated dataset knowing that the events occur in the reference at the same time. It is given by the following formula :

$$\mathbf{hit} = \frac{TP}{TP + FP} \tag{10}$$

TP: True positives are events correctly detected by the two datasets at the same time;

FP: False positives are events not detected by the evaluated dataset but that occured in the reference.

Hit values are ranging from 0 to 1; hit=1 means that all the events observed in the evaluated dataset occurred in the reference. Previous work such as (Olauson, 2018; Ramon et al., 2019) have shown that ERA5 provides a good representation of various near surface meteorological variables including near surface humidity and wind speed, in comparison to others reanalyses, including MERRA. Therefore for the computation of the Hit, we choose ERA5 as the reference and MERRA as the evaluated dataset.

#### – ACC

The ACC is similar to a linear correlation, the only difference being that it is calculated using the anomalies of the variables with respect to the climatology. This metric is stricter than the simple correlation and not sensitive to the seasonal cycle which tends to increase the correlation between the products. ACC takes values between -1 and 1. ACC=1 indicates a perfect correlation between the products. For example, to compute the ACC between ERA5 and MERRA using the variable T2m in this study, we firstly compute the anomalies between each reanalysis and their respective climatologies; then we compute the correlation between the resulting anomalies.

  **– GSS**

The GSS, known also as the Equitable Threat Score (ETS), measures the fraction of observed events that are correctly predicted, adjusted for Hits associated with random chance. The GSS does not take in account positive outcomes due to chance. It is stricter than the Hit rate; GSS takes values between $\dfrac{-1}{3}$ and 1. GSS=0 indicates no skill or no correlation while a GSS=1 perfect skill. Given a contingency table (see [Table2]), the computation of the GSS is done by the following formula:

$$\mathbf{GSS} = \frac{\mathbf{A} - \mathbf{CH}}{\mathbf{A} + \mathbf{B} + \mathbf{C} - \mathbf{H}} \tag{11}$$

With CH given by:

$$\mathbf{CH} = \frac{(\mathbf{A} + \mathbf{B})(\mathbf{A} + \mathbf{C})}{\mathbf{A} + \mathbf{B} + \mathbf{C} + \mathbf{D}} \tag{12}$$

## 3  Results

### 3.1  Uncertainties in the reanalysis products

The first step of this work consists in assessing the evolution of $T2m$ in the ERA5 and MERRA reanalyses. The climatological state (annual mean) of $T2m$ in ERA5 and MERRA has been evaluated over the West Africa region from 1 January 1993 to 31 December 2020 [Fig3 (a-b)]. Both reanalyses show very similar climatologies of $T2m$: a north-south gradient of the temperature. The Sahel region appears to be warmer than the Guinean region; this is because of the advection of cold air coming from the Atlantic ocean to the Guinea coast. This fresh air tends to cool down temperatures in this region. The bias between ERA5 and MERRA is computed using ERA5 as reference [Fig3 c)]. MERRA shows a cold bias with respect to ERA5 over the Sahel region and Guinean zone except in some countries (e.g., Guinea Bissau, Sierra Leone, Liberia) where we observe a hot bias. The Biases between ERA5 and MERRA are around +/-2°C. The bias highlighted between ERA5 and MERRA is very significant for heat wave detection. Thereafter, we evaluate the temporal consistency between the two reanalyses by computing the ACC for $T2m, AT$ and $Tw$ (see [Fig3 (d-i)). We observed a weak correlation over the south of Sahel and Guinean region around 0.5 (0.7) for maximum (minimum) values of $T2m$ and $AT$ (see [Fig3 (d-e) and (g-h)]). This could be explained by the presence of a strong diurnal cycle in the region associated with high variability during the day and less variability during the night. This will lead to a high variability in the daily maximum values compared to the daily minimum values. $Tw$ shows a uniform repartition of correlation between ERA5 and MERRA around 0.85 except in the Guinean region with maximum values. Good agreement between the 2 products is found with $Tw$. We can infer from this result that $Tw$ has a more stable signal than $T2m$ and $AT$. Knowing that heat waves are defined as extreme events, it is important to evaluate the consistency of the reanalysis products on the representation of extreme values. The Hit rate and GSS were calculated in terms of hot days using $T2m$, and we noticed very low values between the two reanalysis products over the southern Sahel and Guinean region around 0.25 (see [FigS4] in the supplemental material). Similar results have been found with $Tw$ (not shown). The lack of coherence between ERA5 and MERRA on the representation of hot days would result in discrepancies

on the number of heat wave events derived from the two reanalyses. The analysis of heat waves frequency in the two products using $T2m$ and $AT$ shows big differences over the coastal region (see [FigS5] in the supplemental material). This is very consistent with the ACC results shown earlier. These discrepancies in the reanalyses ERA5 and MERRA in West Africa were also highlighted by Engdaw et al. (2022). The potential origins of these differences are explored in the discussion section. The spatial variability of heat waves occurrence in ERA5, using $T2m$ and $AT$ as indicators, is very similar regardless of the

methods applied for heat waves detection. This strong correlation between T2m and AT is also observed when using MERRA reanalysis (see [FigS6]). Even if the reanalyses show discrepancies over the south of Sahel and coastal region with respect to key variables, the correlation between the variables is preserved.

### 3.2  Sensitivity of heat wave detection to threshold values

As discussed earlier in the section "heat wave detection", the threshold value used for heat waves monitoring has a significant

impact on heat wave characteristics. The threshold value is generally tailored to the application that is to be carried out. In this part of the work, we investigate the sensitivity of heat waves freaquency on different thresholds. To achieve this goal, we define 4 relative threshold values calculated over the entire period: the $75^{th}, 80^{th}, 85^{th}$ and $90^{th}$ daily percentiles. The choice of these thresholds for assessing changes in heat wave characteristics is based on previous work. Many studies are using the $90^{th}$ percentile to define a heat wave (e.g., Fischer and Schär, 2010; Perkins et al., 2012a, b; Déqué et al., 2017; Lavaysse et al., 2018;

Barbier et al., 2018); other studies are using the $75^{th}$ percentile (Guigma et al., 2020). Based on these studies, we decided to test the sensitivity of threshold values from the third quartile ($75^{th}$) to $90^{th}$ percentile by steps of 5% to quantify significant changes in heat waves frequency. As we are studying extreme events, it is not relevant to go below the third quartile; knowing also that this study focuses on human impacts of heat waves, the $90^{th}$ percentile is enough as a maximum threshold. Heat waves detection is treated separately for these 4 thresholds (see [FigS7] in the supplemental material). The sensitivity of heat waves frequency

or duration with respect to the thresholds ($75^{th}, 80^{th}, 85^{th}, 90^{th}$ percentiles) is treated independently for the 4 thresholds; this is done by calculating the linear evolution coefficient over each grid point. The linear evolution coefficient is defined as the slope of the linear regression line fitted between the threshold values ($Q_{75}, Q_{80}, Q_{85}, Q_{90}$) and the number of events associated to each threshold ($NQ_{75}, NQ_{80}, NQ_{85}, NQ_{90}$) or their corresponding duration ($DQ_{75}, DQ_{80}, DQ_{85}, DQ_{90}$). The calculation of the linear evolution coefficient is carried out according to the following steps:

– After processing to heat waves detection at each grid point for the 4 thresholds separately, we compute for each of them the frequency and duration of heat waves;

  – then fitted a regression line between the threshold values ($Q_{75}, Q_{80}, Q_{85}, Q_{90}$) and their corresponding frequency or duration. This is done for each grid point;

  – Finally, the changes in heat waves occurrence/duration from the $75^{th}$ to $90^{th}$ percentiles at each grid point, is given by

the computation of the slope of the regression line fitted at step 2 between the threshold values and their corresponding heat waves occurrence/duration.

We are aware that this regression based on 4 points is not very robust, nevertheless it makes it possible to obtain information on the evolution of the heat wave characteristics with respect to the thresholds. We therefore assessed the significance of the slope values with respect to the thresholds using a confidence level of 95%. The significance of the slope was evaluated using a two sided Chi-square statistics test (Pandis, 2016).

The linear evolution is given by the following equations:

$$\mathbf{N = a_w * threshold + b_w} \tag{13}$$

$$\mathbf{D = a_w^{'} * threshold + b_w^{'}} \tag{14}$$

where $a_w, a_w^{'}$ and $b_w, b_w^{'}$ are respectively the slopes and intercepts of the regressions at the grid point w for heat waves frequency and duration.

This analysis is conducted with $T2m$ and $Tw$ extracted from MERRA and ERA5 reanalyses. We find a high spatial variability in the sensitivity of heat waves occurrence to the threshold values over West African regions ([FigS8] and [FigS9] in the supplemental material). Some regions are more sensitive than the others; this can be explained by a strong seasonal cycle of the $T2m$ and $Tw$ signals in those regions. We observe small changes in the frequency and duration of heat waves with respect to the thresholds when using the minimum and maximum values of $T2m$ or $Tw$ ([FigS8 (c,f)], [Fig4 (c,f)] and [FigS9 (c,f)] in supplement material); this is related to the small sample size of events detected with method 3 (see section heat wave detection for more details). As the results show, the frequency of heat waves can be expected to increase with decreasing threshold values (see [FigS8] and [FigS7] in the supplemental material). Heat waves detected using low threshold values are very persistent and last for several days ([Fig4] shows an illustration with $T2m$ used as indicator). This can be explained by the fact that when using a low threshold value, one can expect to have many days with temperature values above the daily threshold. Conversely, for heat waves detected with high threshold values, the duration of the events is considerably reduced. This is statistically coherent because the number of consecutive days with temperature above the threshold will decrease as the threshold increases. In general, we find that the duration of heat waves is more sensitive to the threshold values than their frequency. This is very coherent because the persistence of a heat wave will be mainly affected by the threshold values used for the detection.

## 3.3 Sensitivity of heat wave detection to the choice of indicators and methods applied

We have shown that the heat waves detection is very sensitive to threshold values. Based on the literature review and the application of this work, for the rest of the study, we use the $90^{th}$ percentile for heat wave analyses (e.g., Fischer and Schär, 2010; Perkins et al., 2012a; Perkins and Alexander, 2013; Fontaine et al., 2013; McGregor et al., 2015; Russo et al., 2016; Mutiibwa et al., 2015; Oueslati et al., 2017; Déqué et al., 2017; Batté et al., 2018; Barbier et al., 2018; Lavaysse et al., 2018; Yu et al., 2021; Engdaw et al., 2022) using ERA5 reanalysis [Fig5]. We identified four indicators: $T2m$, $Tw$, $AT$ and $UTCI$ from which heat waves detection were processed using three different methods (see section methods for more details). We

notice that the occurrences of daytime and nighttime heat waves [Fig5 (a-d);(e-h)] are in the same range of values, while for concomitant events [Fig5 (i-l)], the occurrence of heat waves is drastically reduced by $\frac{1}{4}$. This could be explained by the fact that nighttime and daytime heat waves do not necessarily occur at the same time and their origins are totally different. Daytime heat waves will be mainly influenced by incoming solar radiation, while nighttime heat waves by the water vapor content of the air mass (Barbier et al., 2018; Largeron et al., 2020). We observe a high occurrence of nighttime heat waves over the coastal region from Guinea to Cameroon [Fig5 (a-d)] linked to moist air coming from the Atlantic ocean in the region during the night; daytime heat waves are more frequent on the Sahel and north-east of Sahara [Fig5 (e-h)] due to hot temperatures over the continental regions. When analysing nighttime heat wave events from each indicator [Fig5 (a-d)], it appears that $Tw$ heat waves are more frequent than $T2m/AT/UTCI$ events. It seems that $Tw$ is more sensitive to humidity than the other indicators (see formula of $Tw$); this could explain the high frequency of events observed during the night in the coastal region. Regarding daytime heat waves [Fig5 (e-h)], the spatial variability of events is more consistent for all the indicators in the Sahelian zone. However, some differences are observed: an increase of heat waves occurrence over the coastal region with $Tw$ is noticed compared to $T2m, AT$ and $UTCI$. The detection of heat wave events with method 3, show that $Tw$ events are more frequent than $T2m/AT/UTCI$ events with a maximum of occurrence located over the northern Sahel. This means that daytime and nighttime heat waves occur frequently and simultaneaously over the Sahel with $Tw$. From this result, we can deduce that humidity plays a major role in the occurrence of concomitant heat waves which are very dangerous for human health. In this section, we show the high sensitivity of heat waves detection on the methodology applied and the variables used as indicators. The role of humidity on heat waves occurrence in the coastal region has also been highlighted. Similar results are found with MERRA reanalysis (not shown).

In summary, the heat wave detection is influenced by many parameters: the dataset, threshold values, indicators and methodology used to define such an event. There is a high dependency between these parameters and the climatic region investigated. We illustrate the sensitivity of heat wave characteristics to the previous parameters in the CONT region [Fig6], as well as the ATL and GU regions (see [FigS10] and [FigS11] in the supplemental material) using ERA5 and MERRA reanalyses.

### 3.4 Monitoring of heat waves over West Africa regions

In this section, we analyse the spatial variability of heat waves in three climate regions (CONT, ATL and GU see section "region of interest" for more details) using $T2m$, $AT$, $Tw$ as indicators and reanalysis data. Although ERA5 is slightly better than MERRA when compared to station data [FigS3b], we have evaluated the recent evolution of heat waves in both reanalyses. To do so, we firstly assessed the interannual variability of heat waves and their characteristics from 1993 to 2020. For each region, the characteristics of heat waves were calculated as the ratio of the sum of the characteristics of all the cities belonging to a region divided by the number of cities. We identified some particular hot years with a high frequency of nighttime, daytime and concomitant heat waves: 1998, 2005, 2010, 2016 2019 and 2020 in the 3 regions for all the indicators (see [Fig7], [FigS12 and FigS13] in the supplemental material). These peak heat waves years are addressed in the discussion section. The GU region appears to have experienced more heat waves over the last decade than the CONT and ATL regions ( see [FigS12] for

daytime events and [FigS13] for nighttime in the supplemental material). The mean duration of heat waves detected in the three regions are in the same range of values with some specific persistent events at the end of the period in the ATL and GU regions (not shown). Stronger and more persistent heat waves are found in the CONT region. From a statistical point of view, this is due to less variability in the signal of indicators in the region, which favours the detection of consecutive days with indicator values above the threshold. The highest occurences of heat waves in the three regions are associated with $Tw$ for daytime and nighttime events (see [FigS12] and [FigS13] respectively). Conversely, high intensity heat waves are associated with $AT$ (not shown) in the three regions. We can infer from this result that $AT$ presents a more stable signal in the regions than $T2m$ and $Tw$. Concomitant high intensity events are found in the CONT and ATL regions (see [FigS15] in the supplemental material).

We also investigate the seasonal distribution of heat waves occurrence in the 3 regions. We find an increase in the frequency of daytime and nighttime heat wavess at the beginning of the season and during the retreat period of the West Africa monsoon (starting in September, see [FigS14 (a-c)] in supplement material). A decrease in heat wave frequency is observed during the active phase of the monsoon in the 3 regions; this is consistent because the monsoon flow bringing rainfall in the region, resulting in a cooling effect. The concomitant heat waves show a seasonal cycle with strong fluctuations [FigS16]. This is due to the fact that concomitant events are conditioned by daytime and nighttime heat waves which are two distinct processes.

The seasonal cycle of the duration and intensity of heat waves follows the same distribution as the heat wave occurrence (see [Fig8 (a-c)] and [Fig8 (d-f)] respectively). Persistent and strong intensity heat waves (nighttime, daytime) occur at the beginning and the end of the season, while short duration and low intensity events are occurring during the monsoon phase ([Fig8 (a-c)], [Fig8 (d-f)]). This is verified for all the 3 indicators despite some discrepancies. The period 1993-2020 is then divided into 3 three decades : [1993-2001], [2002-2011] and [2012-2020]; and we evaluate the contribution of each decade on the heat wave characteristics over the whole period (see [Fig9 (a,b,c) - (g,h,i)] for heat waves duration). Results are similar when analyzing the intensity of heat waves (not shown). The percentiles used for the detection of heat waves in each decade are computed over the whole period 1993-2020. It is clearly shown with ERA5 that the major contribution on heat wave characteristics over the period is coming from the last decade [Fig9 (g,h,i)]. We notice a progressive increase in frequency, duration and intensity of all the heat waves (daytime, nighttime and concomitant) from the first to the last decade in the 3 regions (see [FigS14(j-l)] and [FigS16 (j-l)] in the supplemental material); this is true for all the indicators. Using ERA5 reanalysis, we found the last decade [2012-2020] shows a major contribution around 50% of heat wave characteristics over the period 1993-2010, while the first and second decades contribute up to 22.4% and 27.6% respectively. This contribution of the last decade over the total period is not effective in MERRA reanalysis where the three decades appear to have a similar contribution. This is the result of the uncertainties highlighted earlier in both reanalyses. The reinforcement of extreme events such as heat waves during the last decade in ERA5 is possibly being linked to global warming. This result is consistent with other studies that show an increase of heat waves frequency and property under climate change (e.g., Dosio, 2017; Dosio et al., 2018; Murari and Ghosh, 2019; Lorenzo et al., 2021; Engdaw et al., 2022). When analysing the severity of heat waves over the previous decades using the mean duration and intensity (see [FigS17] and [FigS18]), we do not find a significant increase of heat wave characteristics over the 3 decades.

After the evaluation of the temporal evolution of heat waves over the 3 regions, we analyse their persistence based on their duration using ERA5 and MERRA reanalyses and maximum values of the indicators (see [Fig10] and [FigS19] respectively). We defined 5 types of events as described in [Table3]). We observed that approximately 75% of daytime heat waves have a duration of 3-6 days with at least 40% of events belonging to C1 [Fig10]. Very persistent daytime heat waves contribute to at least 9-13% of the events registered. Severe and very severe daytime events are extremely rare in the region; and they contribute up to 12% of the total number of heat waves. The classification is not too sensitive to indicators and regions. We obtained a similar classification with nighttime heat waves (not shown).

## 4  Discussion on the uncertainties found in reanalyses and the impacts of the SST in the Atlantic

We analyze the evolution of heat waves occurrence and characteristics over a variety of climatic regions in West Africa. The spatial variability of heat wave indicators ($T2m$ and $Tw$) over West Africa during the seasons (winter, spring, summer and autumn) was investigated. This is done through the computation of the interannual daily standard deviation over the period 1993-2020. We find the lowest values of standard deviation over the 3 regions of interest (CONT, ATL and GU) during the summer and autumn when we use mininimum values of $T2m(T2m_{min})$ (see [FigS20] in the supplement material). This shows low variability in the signal of $T2m_{min}$ which indicates favorable conditions for the occurrence of persistent heat waves in these regions during this period. With $Tw$, there is a low variability of the signal during the summer for both minimum and maximum values indicating persistent events. We find some discrepancies in the reanalysis products ERA5 and MERRA. The results show that ERA5 appears to be hotter than MERRA over the Sahel region. The source of these discrepancies in the reanalyses is very complex and may result from different factors such as the data assimilation techniques ($4D - Var$ (Bonavita et al., 2016) for ERA5 and $3D - Var$ (Courtier et al., 1998) for MERRA), atmospheric models, convective schemes, bias correction methods, spatial resolution and model parameterization. Another major difference between ERA5 and MERRA is in the vertical resolution of the profiles of the atmospheric variables between 0-2 km; ERA5 has more atmospheric vertical levels than MERRA below 2 km which leads to a more accurate representation of processes in the boundary layer (Taszarek et al., 2021). Many studies highlighted these differences in the 2 reanalysis products (e.g., Olauson, 2018; Graham et al., 2019; Taszarek et al., 2021); some authors (e.g., Gensini et al., 2014; Tippett et al., 2014; Allen et al., 2015; Taszarek et al., 2018; King and Kennedy, 2019) identified model parameterisation and data assimilation technique as possible causes of biases in reanalyses for low level thermodynamic fields. A more detailed study of the source of these uncertainties is beyond the scope of this paper.

An assessment of the origins of heat waves in the coastal regions of West Africa is discussed. One driver of heat waves over the globe highlighted by many studies, is the "blocking high" (e.g., Charney and DeVore, 1979; Coughlan, 1983; Perkins, 2015). This situation occurs when a high system pressure remains in the same region for a longer period of time than is usually expected. The consequence of this phenomenon is the compression of the air mass at the surface, which leads to an increase

in temperatures in the region. Perkins (2015) also identified soil moisture-atmosphere interactions and large-scale climate dynamics as other drivers of heat waves.

To address the origin of heat waves in the coastal region, we firstly analyzed the interannual variability of the Sea Surface Temperature (SST) over the period 1993-2020 using the ERA5 reanalysis. We computed the mean anomalies of SST with respect to the climatology [FigS21]. A warming over the north-eastern and south-eastern tropical Atlantic ocean is observed

in some years: 1998, 2005, 2008, 2010, 2016, 2019 and 2020. This warming over the tropical Atlantic ocean is affecting all the western Africa coastal region. In comparison to the interannual variability of heat waves occurrence in the coastal region (see [FigS12] in the supplemental material), we noticed that the years of high heat waves frequency correspond to years in which ocean warming was observed : 1998, 2005, 2010, 2016 , 2019 and 2020 for instance. These years also correspond to the occurrence of El nino events. The link between the SST and heat waves has been investigated in more detail in the following.

We computed the yearly mean SST anomalies with respect to heat waves days using the formula below :

$$\mathbf{Ano\_SST\_year} = \sum_{i=1}^{12} \alpha_i * \mathbf{Ano_i} = \alpha_1 * \mathbf{Ano_1} + \alpha_2 * \mathbf{Ano_2} + \alpha_3 * \mathbf{Ano_3} + ... + \alpha_{12} * \mathbf{Ano_{12}} \tag{15}$$

Where $\alpha_i$ represents the total number of days in heat waves per month for each year, if there is no event detected then $\alpha_i = 0$.

For this analysis, we focused on the years with high peaks of heat waves identified previously (1998, 2005, 2008, 2010, 2016, 2019 and 2020) using $T2m$ as indicator for heat waves detection. We noticed that most of the heat waves are associated

with a warming of the tropical Atlantic ocean except for some specific years such as 2016 and 2019 in the GU and ATL regions respectively [Figure11 (a,b)]. In the CONT region, heat waves are influenced by both the west-east air flow coming from the tropical north Atlantic ocean and the south-north air flow coming from the tropical south Atlantic ocean [Figure11 (c,d)]. Some years in which a considerable number of heat waves have been detected are not associated with positive anomalies of SST. These heat waves are occurring during a cold phase of the tropical Atlantic ocean. There is no major changes observed in

the analysis when using $AT$ and $Tw$ as indicators for heat waves detection (not shown). We can suggest from this result that heat waves in the coastal region have many drivers, and one of them at local scale could be the oceanic forcings through the SST. Large scale (El nino, atmospheric circulation) and local scale (soil-moisture interactions) processes may also contribute to the occurrence of heat waves in the region. This result is in agreement with Russo et al. (2016) and Moron et al. (2016) who identified links between heat waves and El niño events. The investigation of the physical processes driving heat waves in the

coastal region required more in-depth knowledge of local and large-scale forcings, which is beyond the scope of this paper.

## 5 Conclusions

The present work assesses the potential uncertainties associated with heat waves detection using reanalysis data (ERA5, MERRA). It also looks into the recent evolution of heat waves in different parts of the West African region.

The first uncertainty highlighted in this study comes from the ERA5 and MERRA reanalyses. We found biases in the

510 reanalysis products; MERRA shows a cold bias compared to ERA5 over the Sahel region and the Guinean region except

over some countries (Guinea Bissau, Sierra Leone, Liberia). Weak correlations between ERA5 and MERRA were found over the Guinea coast using minimum/maximum values of $T2m$ and $AT$ indicators. The representation of extreme values in the reanalyses was analyzed, showing that the coherence between the 2 products is very low, around 0.25, in the southern Sahel and the Guinean region. This low agreement between the two reanalyses results in discrepancies in the frequency of heat waves associated with each product. Even though the reanalyses present large discrepancies over the southern Sahel and Guinea coast, they are able to preserve the relationship between the variables used for heat waves detection ($AT$, $T2m$). Temperatures estimated from local station in Dakar and Abidjan show slightly better correlation with ERA5 than MERRA. The second uncertainty found here, is the sensitivity of the spatial variability of heat waves to the threshold values used to process the monitoring of events. Heat waves detected using low threshold values of the indicators $T2m$ and $Tw$ are very persistent and last for several days, while the duration of heat waves related to the high threshold values is considerably reduced. We notice some discrepancies in the sensitivity to the threshold values of heat waves detected with $Tw$ and $T2m$. Nighttime and daytime heat waves are in the same range of occurrence, while concomitant events are extremely rare because they are more restrictive. This shows that daytime and nighttime heat waves are distinct phenomena. The climatological state of heat waves occurrence shows large differences between the indicators. Nighttime heat waves associated with $Tw$ are more frequent than those detected with $AT$, $T2m$, $UTCI$. This shows that humidity plays an important role in nighttime events and tends to reinforce concomitant events over northern Sahel. The spatial variability of daytime heat waves is more consistent for all indicators over the Sahel. The interannual variability of heat waves in the coastal region of West Africa shows for the 3 indicators ($AT$, $T2m$, $Tw$) some particularly hot years with a high frequency of events : $1998, 2005, 2010, 2016, 2019$ and $2020$ linked to El Niño events. The GU region is more affected by heat waves during the last decade (2012-2020) than the CONT and ATL regions. The CONT region experienced more persistent and higher intensity heat waves than the GU region. The seasonal cycle of heat waves shows an increase of the frequency of the events at the beginning of the season and during the retreat phase of the West African monsoon. Conversely, a decrease of heat waves occurrence is observed during the monsoon activity period in the 3 regions. We observed a reinforcement in the frequency, duration and intensity of heat waves during the last decade (2012-2020). This is a consequence of global warming acting on extreme events. No significant changes on the severity of heat waves have been found in the regions during the 3 decades. Most of the events detected in the regions (75% ) have a duration around 3-6 days. The most dangerous events, lasting at least 10 days, accounted for up to 12% of the total number of events. We noticed strong links between SST and heat waves during some specific peak event years, but this is not the case for 2016 and 2019 in the GU and ATL regions, respectively. We can infer from this result that there is a contribution of oceanic forcings in the reinforcement of heat waves in the coastal region among many other drivers. In a future work, we will investigate in more detail the influence of large-scale forcings on heat waves occurrence in this region. In the present study, we detected different types of heat waves based on the methodology and indicator used, it will be very important to investigate their potential impacts on human health and activities.

*Acknowledgements.* This work is supported by the French National Research Agency in the framework of the STEWARd project under grant ANR-19-CE03-0012 (2020-2024).

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

## Tables

**Table 1.** Land sea mask (lsm) of west African towns used in this study

| Towns | latitude | longitude | lsm |
|---|---|---|---|
| DAKAR | 14.75 | -17.25 | 0.6 |
| ABIDJAN | 5.25 | -3.75 | 0.5 |
| NOUAKCHOTT | 18 | -16 | continent |
| CONAKRY | 9.5 | -13.5 | 0.5 |
| MONROVIA | 6.25 | -10.75 | 0.6 |
| BAMAKO | 12.5 | -8 | continent |
| YAMOUSSOUKRO | 6.75 | -5.25 | continent |
| OUAGADOUGOU | 12.25 | -1.5 | continent |
| ACCRA | 5.5 | -0.5 | 0.8 |
| LOMÉ | 6 | 1 | 0.5 |
| NIAMEY | 13.5 | 2 | continent |
| COTONOU | 6.5 | 2.5 | 0.7 |
| LAGOS | 6.5 | 3.5 | 0.5 |
| ABUJA | 9 | 7.5 | continent |
| DOUALA | 4 | 9.75 | 0.9 |

**Table 2.** Contingency table.

| 2X2 Contingency table | | Event Observed | |
|---|---|---|---|
| | | YES | N0 |
| Event forecast | YES | A | B |
| | NO | C | D |

**Table 3.** Classification of heat waves based on the duration.

| Classes | Duration (days) | Degree of persistence |
|---|---|---|
| C1 | 3 | normal |
| C2 | 4-6 | persistent |
| C3 | 7-9 | very persistent |
| C4 | 10-12 | severe |
| C5 | +13 | very severe |

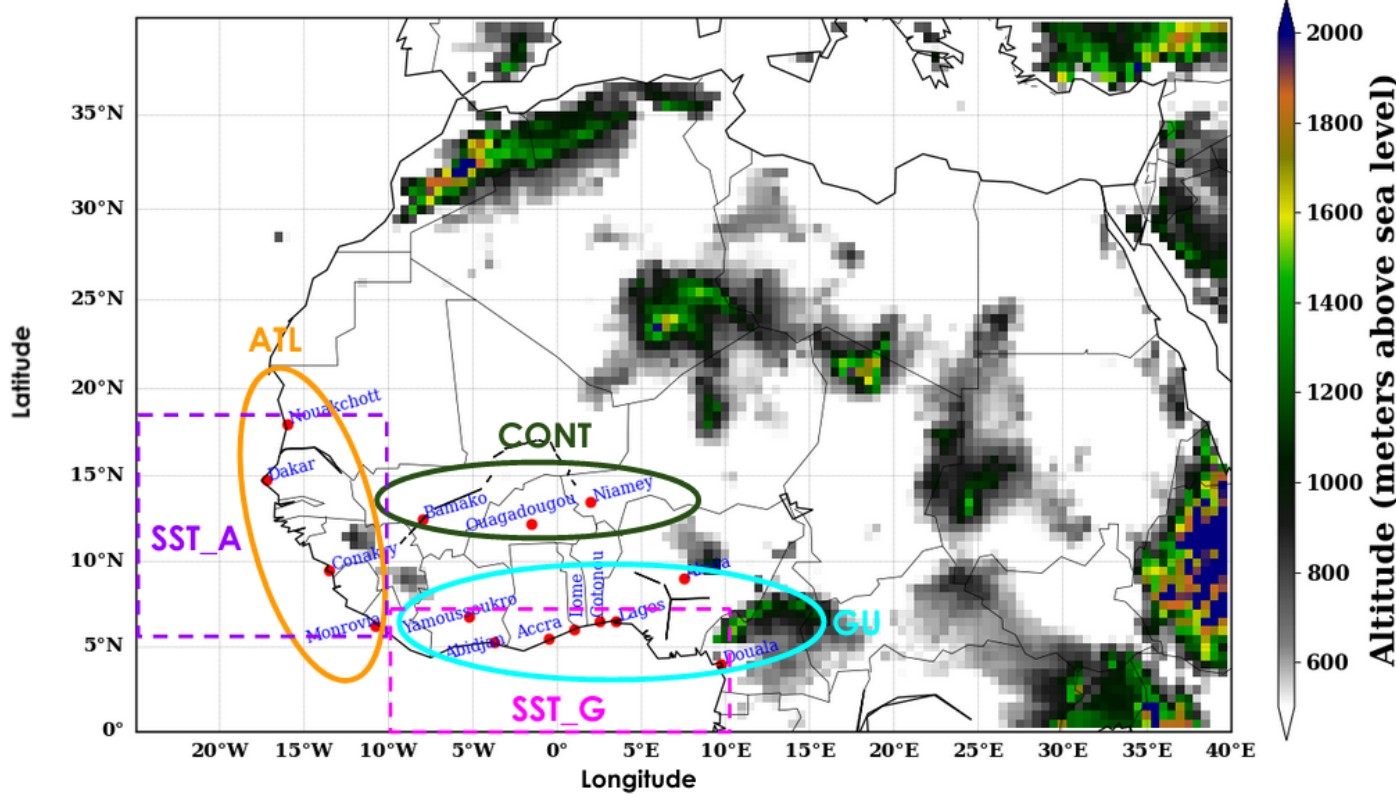

**Figure 1.** *Topographic map of West Africa using ERA5 elevation data. The circles on the map represent the different climatic zones: ATL (Coastal atlantic zone), CONT (Continental zone) and GU (Coastal Guinean zone). The two boxes namely SST_A and SST_G represent the boxes used to analyze the links between Sea Surface Temperature (SST) and heat waves for the ATL and GU regions respectively. The Y- and X- axis represent the latitude and longitude respectively. The color bar shows the elevation in meters over the region.*

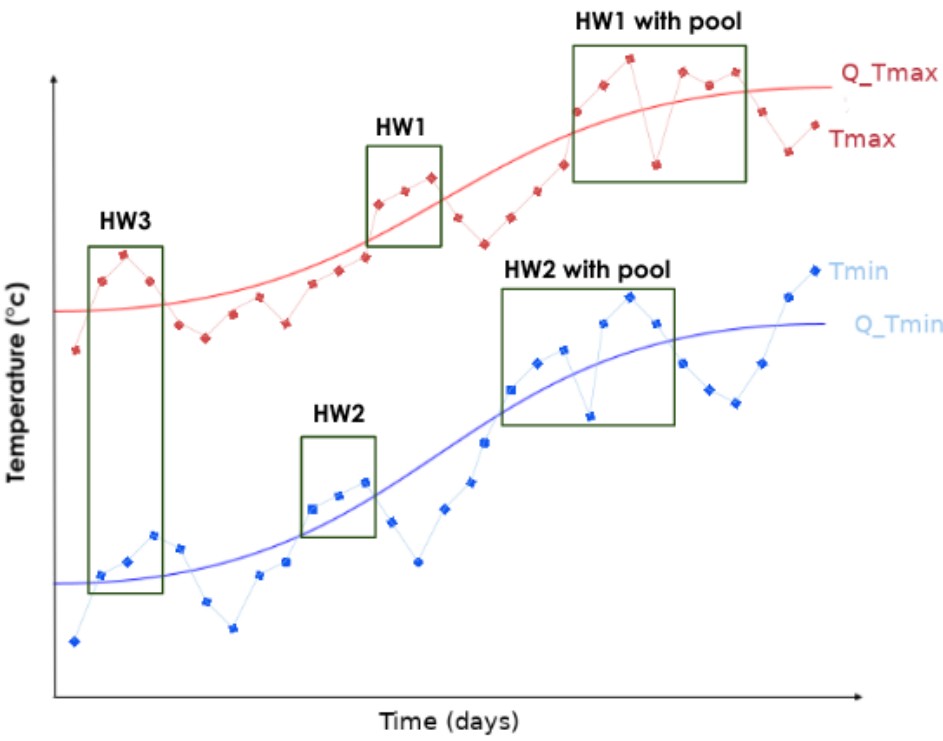

**Figure 2.** *Schematic illustration of heat wave detection process:* **HW1/HW2** *represent heat waves associated to maxima/minima temperature,* **HW3** *are heat waves detected at the same time in maxima and minima temperatures. The red/blue lines with circles are max/min daily temperatures. Red/blue solid lines are max/min thresholds. X- and Y- axis represent the time in days and the temperature in degrees Celsius respectively. 'With pool' refers to the pooling of two (or more) events separated by one day below the daily threshold. This figure shows the different types of heat waves investigated in this work.*

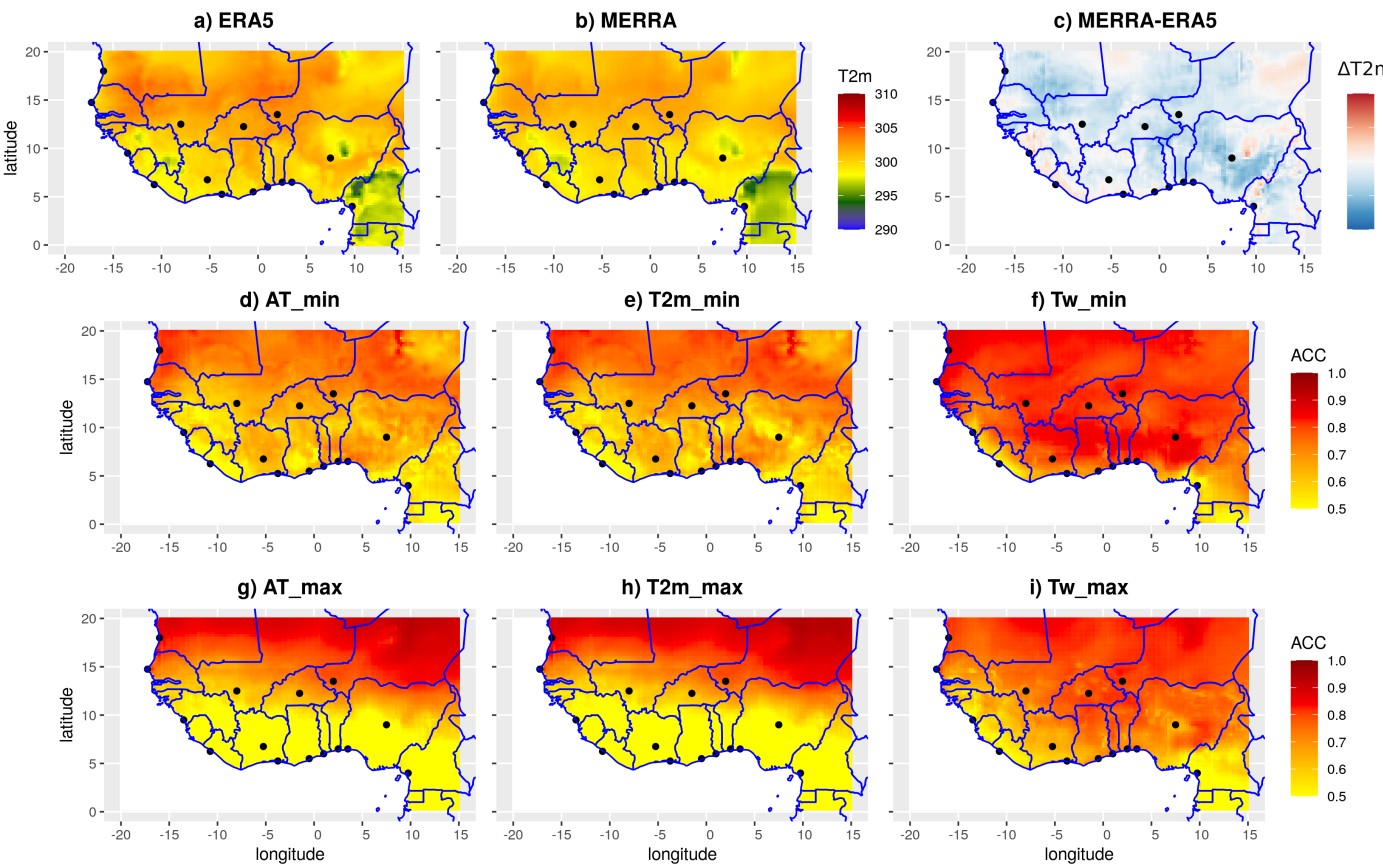

**Figure 3.** *Assessment of the evolution of some atmospheric variables in the reanalyses data: **a-b)** Climatology state of T2m over 1993-2020 respectively for ERA5 and MERRA ; **c)** Climatological bias between MERRA and ERA5 using ERA5 as reference ($\Delta T2m$). **d-f) / g-i)** Anomaly of correlation between MERRA and ERA5 respectively for min / max values using AT, T2m and Tw variables. X- and Y-axis represent the longitude and latitude in degrees respectively. The color bars show the temperature (T2m) in degree Kelvin and the values of anomaly of temperature (ACC) respectively. The black points in the map represent the cities of interest analysed in this work (see section region of interest for more details), this apply for all the maps in the paper.*

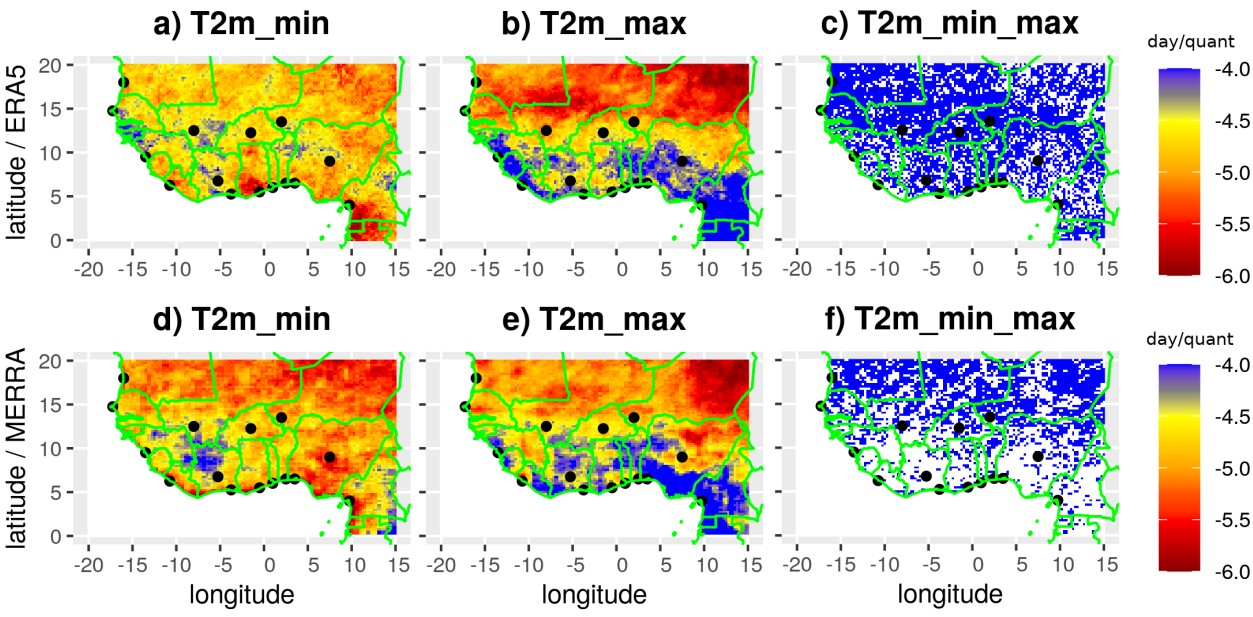

**Figure 4.** *Evolution of the heat wave duration with respect to the threshold values using T2m as indicator for : **a-c)** ERA5 and **d-f)** MERRA respectively. The figure shows the slope of the regression line in day per percentile which is computed by fitting a linear regression between the threshold values (Q75, Q80, Q85, Q90) and their corresponding heat waves's duration ($DQ_{75}, DQ_{80}, DQ_{85}, DQ_{90}$).* X- and Y- axis represent the longitude and latitude in degrees respectively. The color bar shows the values of the slope. The white blanks indicate non significant changes in the duration of heat waves per percentile. The significance of the slope of the regression line has been computed using a two-sided Chi-square test.

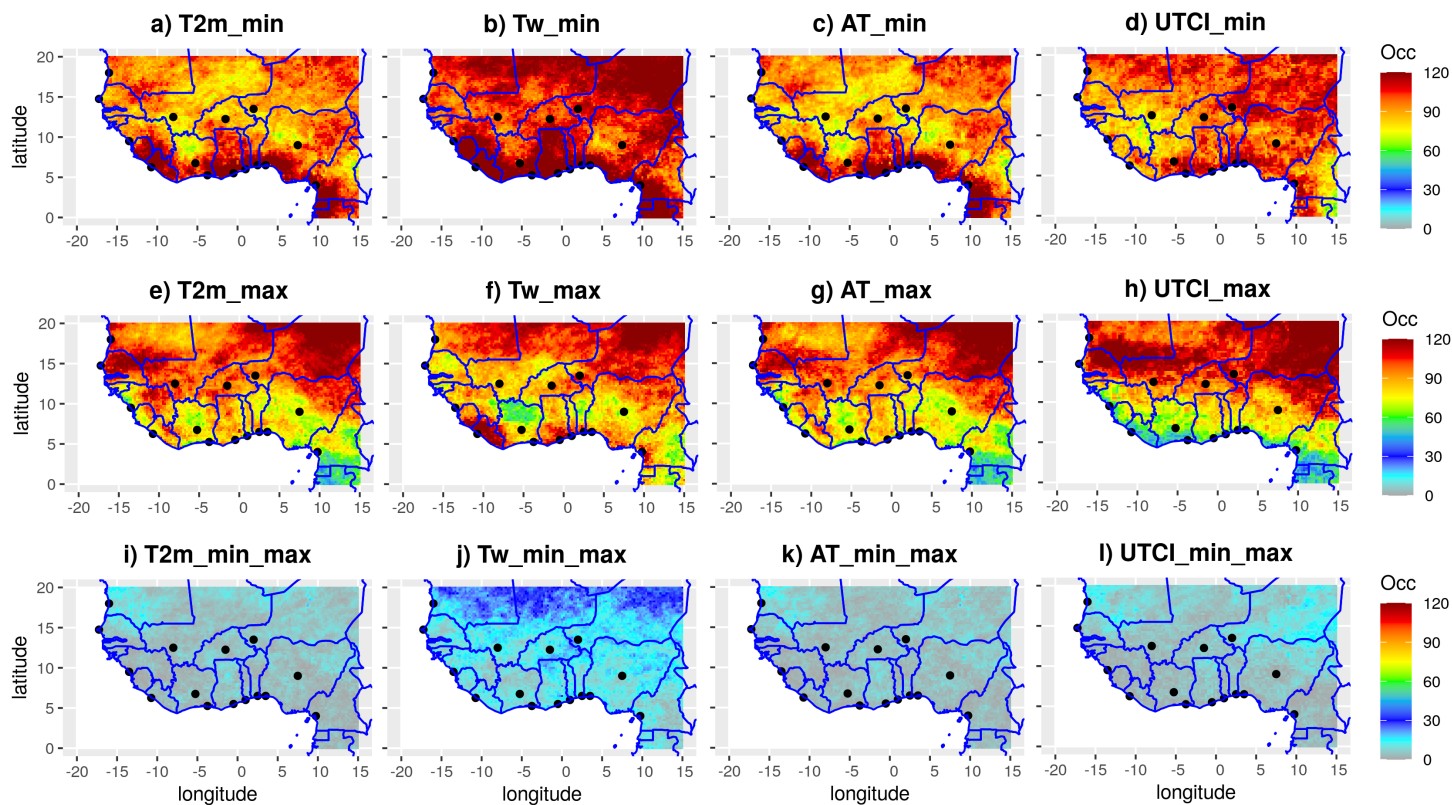

**Figure 5.** *Climatological state of heat waves occurrence over West Africa during the period 1993-2020 using four different indicators ($T2m, Tw, AT, UTCI$). The detection of heat waves is based on the definition adopted : **a-d)** minimum values of indicators, **e-h)** maximum values of indicators and **i-l)** minimum and maximum values of indicators. The detection of heat waves was processed using ERA5 reanalysis and the climatological daily $90^{th}$ percentile over the period as threshold. X- and Y- axis represent the longitude and latitude in degrees respectively. The color bar shows the frequency of heat waves per region.*

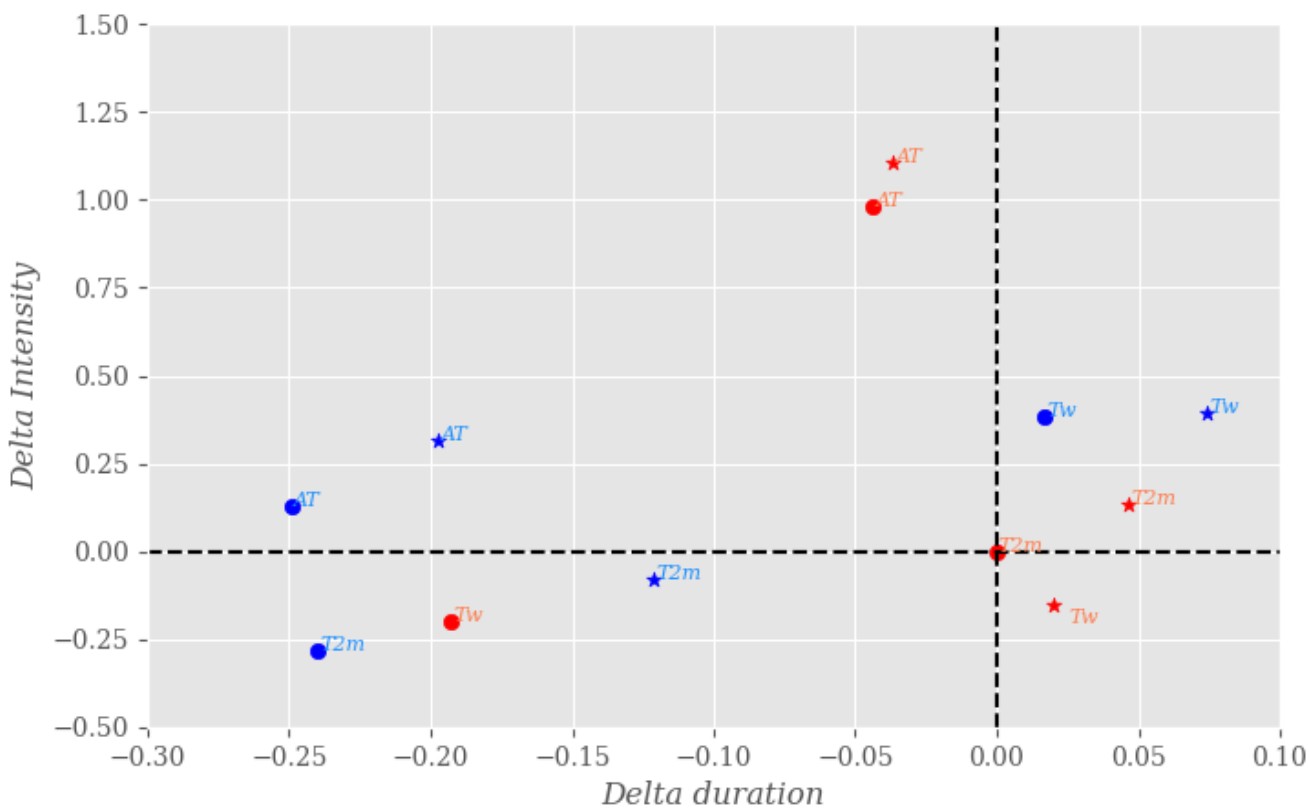

**Figure 6.** *Sensitivity analysis of heat wave characteristics to the datasets, indicators and methodologies used in the CONT region. The characteristics investigated here are the duration and intensity. The circles and stars in the figure represent ERA5 and MERRA reanalyses respectively. The blue/red color represents minimum/maximum values of the indicators. "$T2m_{max}$" from ERA5 is the reference variable used for this analysis. The Y- and X- axis show the standardized variation of intensity and duration from the reference (no unit) respectively. The variation of duration and intensity have been computed using max daily T2m in ERA5 as reference. The detection of heat waves is done using the climatological daily $90^{th}$ percentile over the period as threshold.*

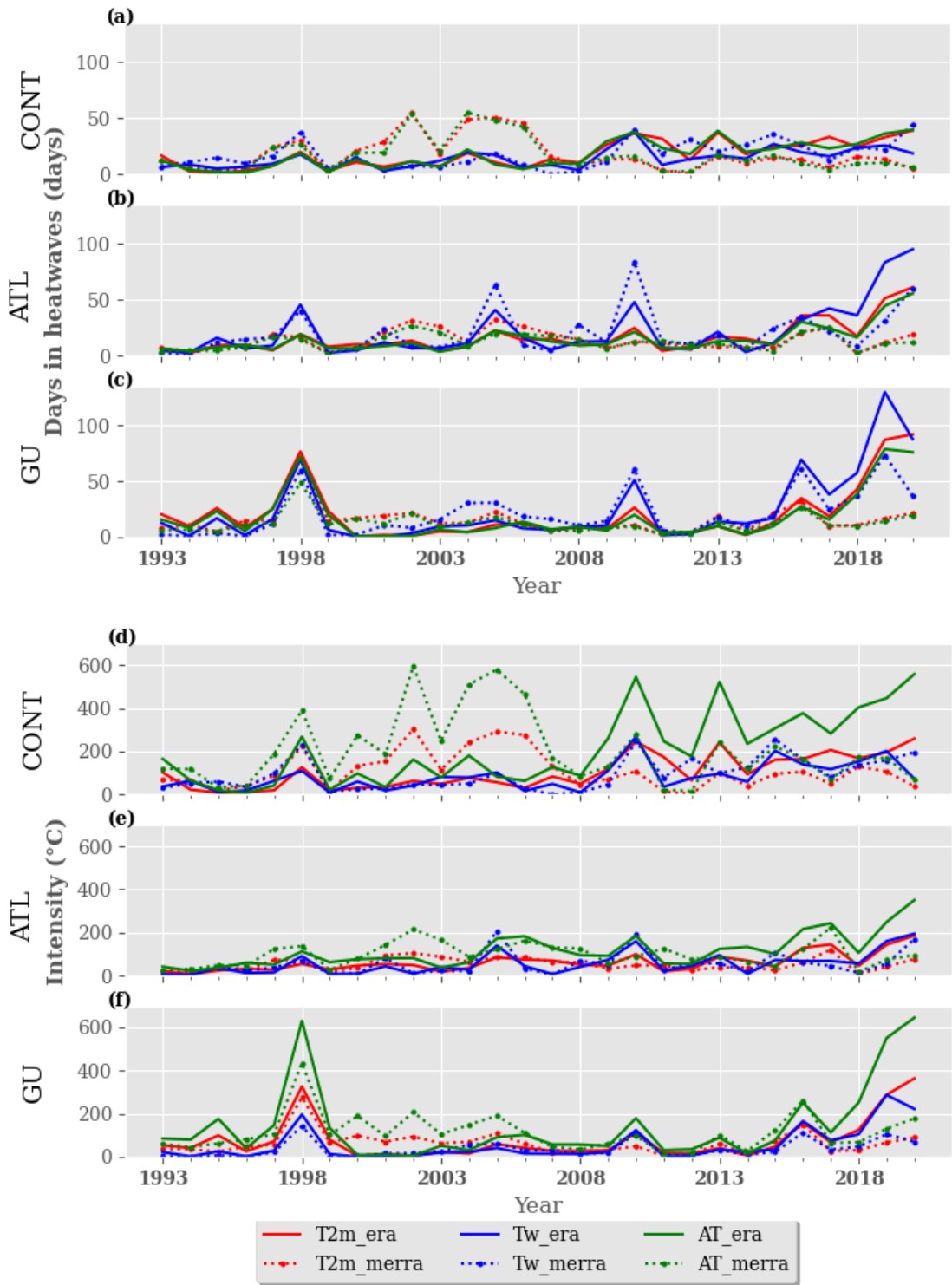

**Figure 7.** *Interannual variability of heat waves characteristics using maximum values of $T2m, Tw$ and $AT$ for : **a-c)** duration and **d-f)** intensity respectively. The detection of heat waves is done using the climatological daily $90^{th}$ percentile as threshold over the period; and the characteristics of heat waves are computed in the 3 regions: **a,d)** $CONT$, **b,e)** $ATL$, **c,f)** $GU$ respectively. The Red/blue/green strong and dashed lines represent the evolution of heat waves characteristics using $T2m, Tw, AT$ from ERA5 and MERRA respectively. The Y- and X- axis represent the duration and intensity of heat waves and the time in year respectively.*

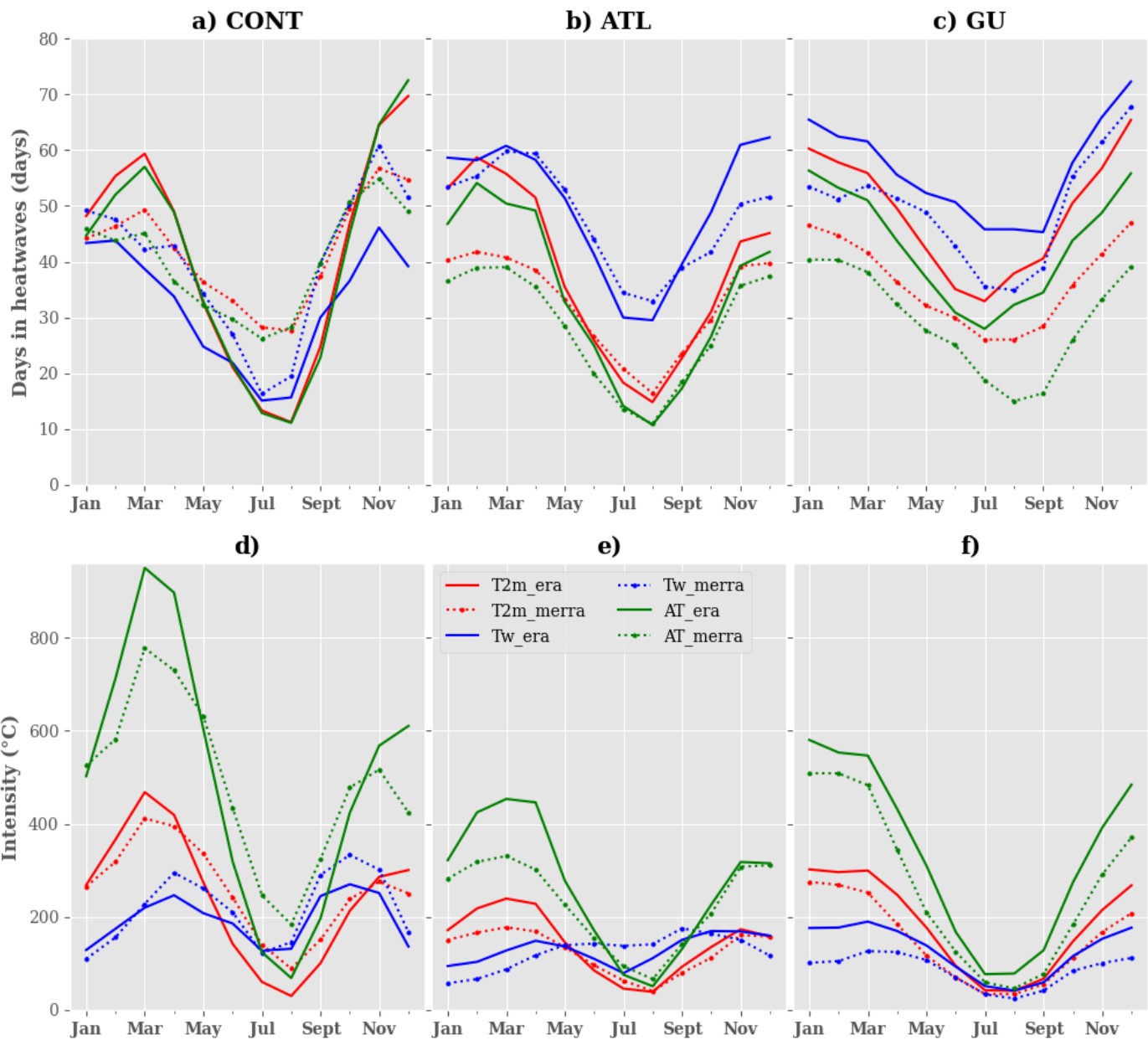

**Figure 8.** *Seasonal variability of heat waves characteristics using maximum values of $T2m, Tw, AT$ for : **a-c)** duration and **d-f)** intensity respectively. We compute a 3-month running mean to smooth the seasonal cycle. The detection of heat waves is done using the climatological daily $90^{th}$ percentile as threshold in the different regions : **a-d)** CONT , **b-e)** ATL, **c-f)** GU respectively. The Red/blue/green strong and dashed lines represent the evolution of heat waves characteristics using $T2m, Tw, AT$ from ERA5 and MERRA respectively. The Y- and X-axis represent the duration and intensity of heat waves and the time in month respectively.*

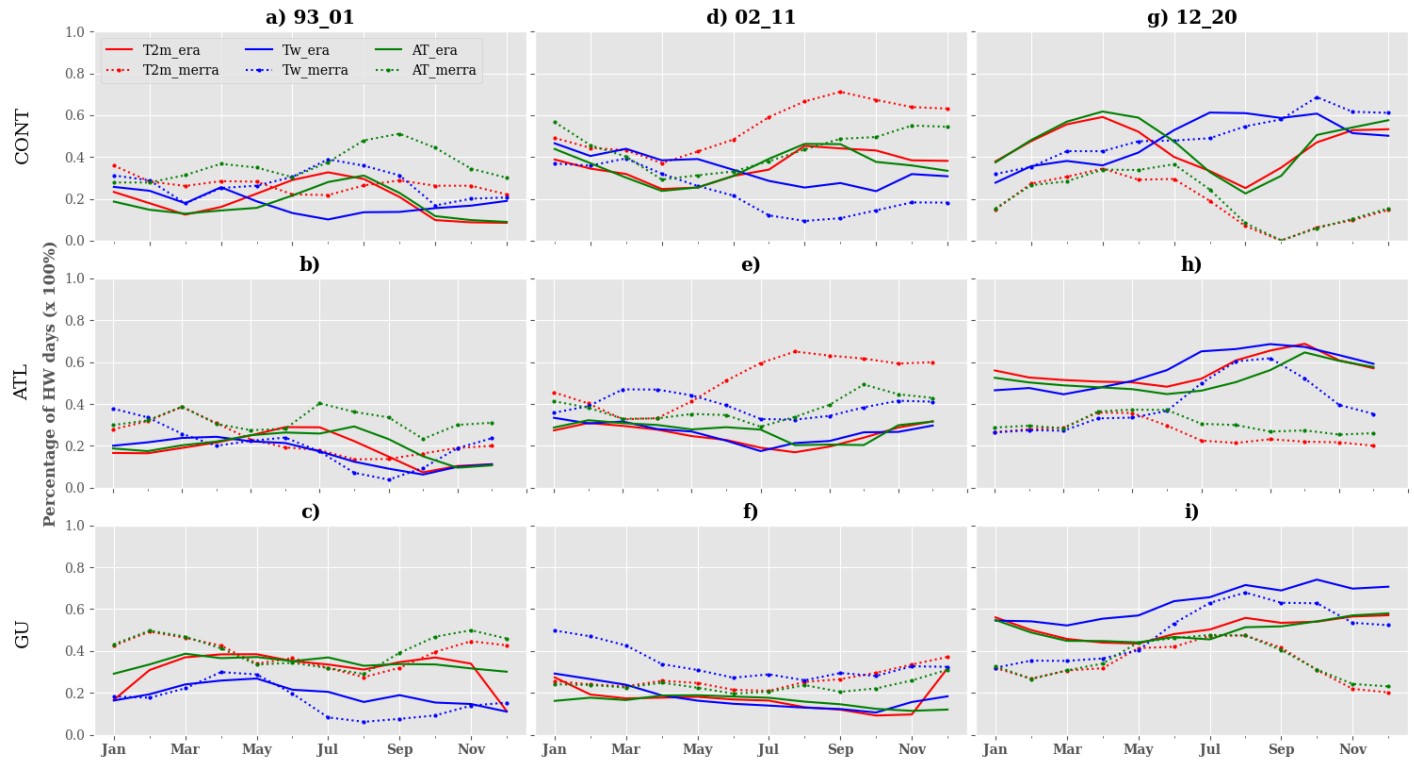

**Figure 9.** *Contribution in percentage of the different decades to heat wave duration using maximum values of $T2m, Tw, AT$ over the whole period respectively for : **a-c)** 1993-2001, **d-f)** 2002-2011 and **g-i)** 2012-2020. We compute a 3-month running mean to smooth the seasonal cycle. The detection of heat waves is done using the $90^{th}$ percentile as threshold over the period in the different regions : **a,d,g)** CONT, **b,e,h)** ATL and **c,f,i)** GU respectively. Red/blue/green strong and dashed lines represent the evolution of heat waves duration using $T2m, Tw, AT$ from ERA5 and MERRA respectively. The Y- and X- axis represent the percentage of heat wave days and the time in month respectively.*

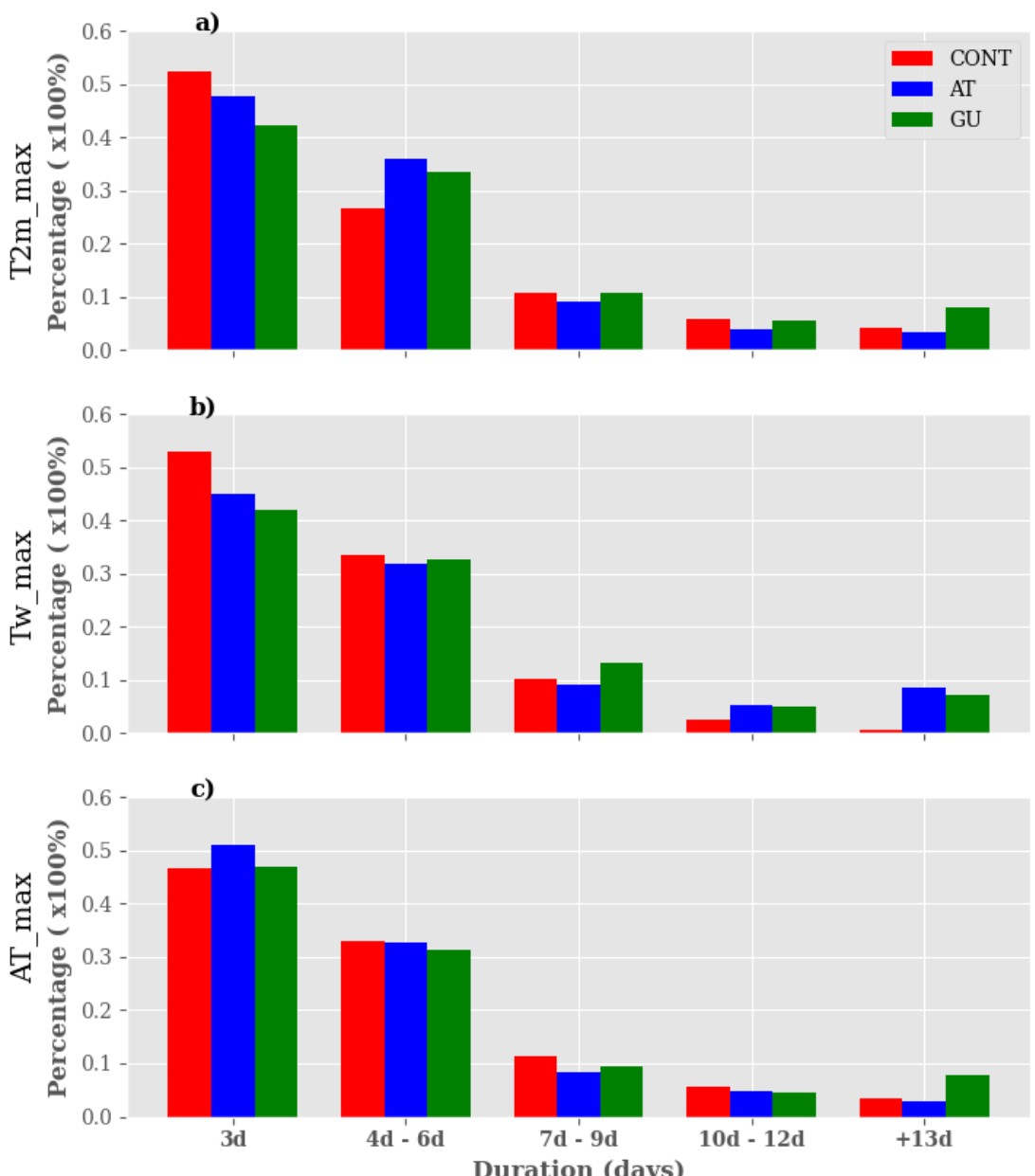

**Figure 10.** *Classification of heat waves using ERA5 reanalysis based on their persistence over the period 1993-2020 : **a)** T2m, **b)** Tw and **c)** AT. The detection of heat waves is done using maximum values of the indicators and the climatological daily $90^{th}$ percentile. Heat waves detection is firstly proceed and then their duration is computed. Clusters of heat waves based on their duration (3d, 4d-6d, 7d-9d, 10d-12d, +13d) are created and finally, we quantify the proportion of each class of heat waves to the total number of events detected.* The Y- and X- axis represent the percentage of the heat waves per class and the duration in day respectively. The Red/blue/green bars represent the percentage of heat waves detected over CONT/ATL/GU regions respectively (see region of interest section for more details). The sum of the contribution of heat waves in different clusters is equal to 1 for each region.

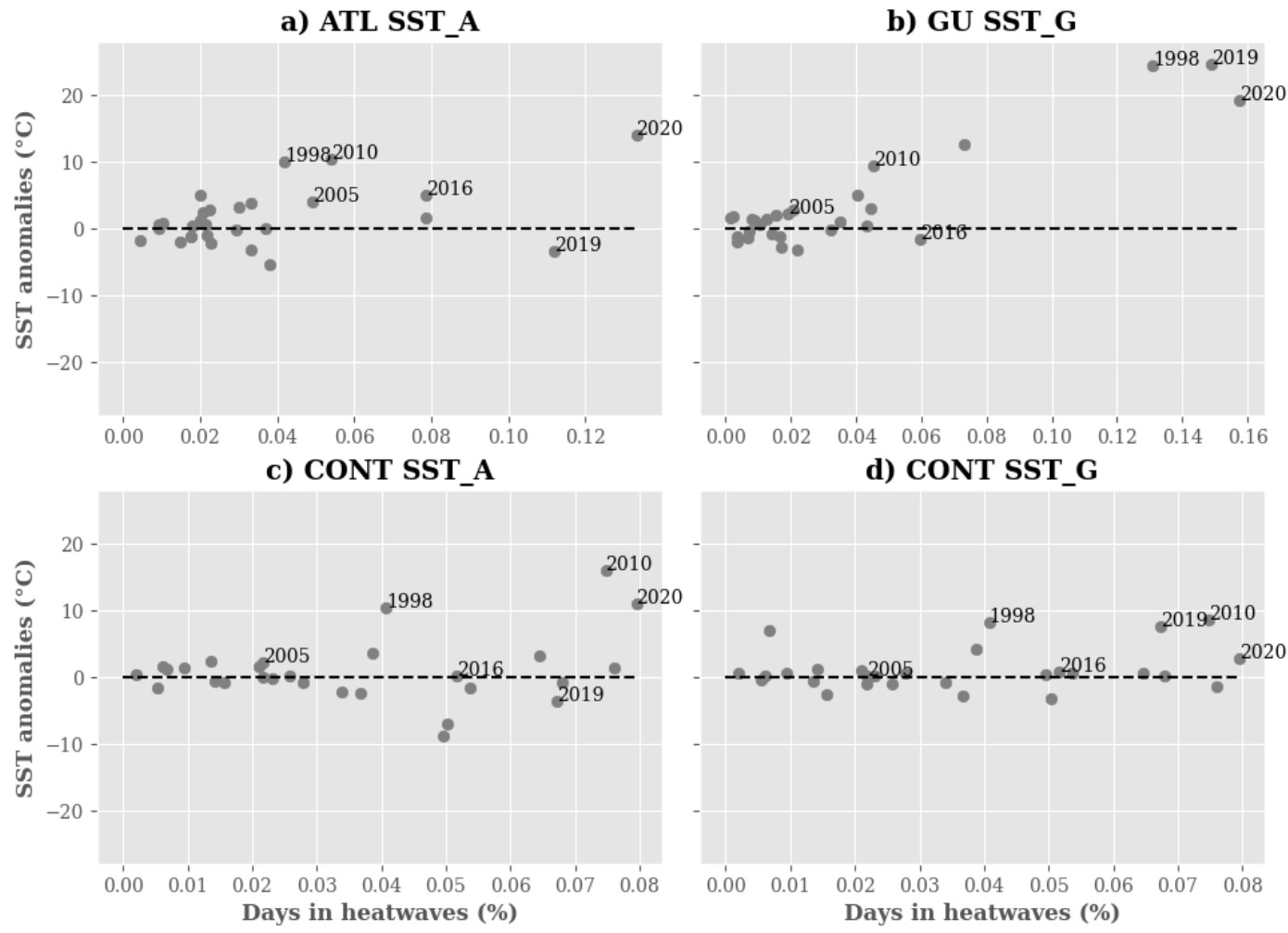

**Figure 11.** *Analysis of the link between SST anomalies and heat waves days over the period 1993-2020 using ERA5 reanalysis and maximum values of $T2m$ in the different regions: **a**) $ATL$, **b**) $GU$ , **c-d**) $CONT$. The "$SST\_A$" and "$SST\_G$" represent the boxes using to compute the SST mean anomalies for the ATL and CONT regions respectively (see [Fig1]). The anomalies are computed as the difference between monthly SST and the monthly climatological values of the SST over the whole period. For each year, the yearly anomalies of SST is computed with respect to the heat waves days. The X- and Y- axis represent the frequency of heat waves days and the SST anomalies respectively. The gray dots represent the years over the period 1993-2020.*