# Peer review of "Heat waves monitoring over West African cities: uncertainties, characterization and recent trends"

_Natural Hazards and Earth System Sciences, 2022_

## Author Comment (AC1)

**Comment on nhess-2022-192**
**Anonymous Referee #1**

Referee comment on "Heat waves monitoring over West African cities: uncertainties,characterization and recent trends" by Cedric Gacial Ngoungue Langue et al., Nat. Hazards

Heat waves monitoring over West African cities: uncertainties, characterization and recent trends

**General comments**

This manuscript assesses potential uncertainties encountered in the process of heat wave monitoring and analyse their recent trend in West African cities. This is investigated using downscaled ERA5 and MERRA variables for the period 1993-2020. Three types of uncertainties are discussed. The first type is related to the reanalyses themselves; the second, to the sensitivity of heat wave frequency to the threshold values used to define them; and, finally, the third is related to the choice of indicators and the methodology used to define heat waves.

We thank the reviewer for his/her availability and interest to examine this work; and also for his/her insightful suggestions to improve the quality of the paper.

**Specific comments**

The abstract is rather long. It contains a surplus of information that could better fit in other sections of the paper such as the introduction. I might suggest keeping it more concise, only stating the main problems & objectives, how they have been dealt with and the main results obtained. The abstract does not mention that local stations have been used nor the downscaling methodology applied. As it is now it seems that it only uses reanalysis data, and this gives a sense of contradiction with the title (which emphasizes the application to cities).
Thanks for this constructive remark, we rearranged the abstract as suggested in the comment.

We changed :

[revised manuscript text omitted]

The analysis conducted to define the three areas is not provided (this is highlighted by the authors). Since this division is a core aspect of the paper, I think it is important to provide, at least, a description of the method followed to obtain these three areas (it can be included as Supplementary Material if the authors consider that it is too dense for the main document).

Thanks for this remark.
We added some description of the method in the manuscript.

We changed:
"The choice of these regions has been validated by conducting some analyses over the cities belonging to each region (not shown). The repartition of the different climatic regions is given as follows :
– Continental zone (CONT hereafter) including the cities of Bamako, Ouagadougou and Niamey [Fig1];
– Coastal atlantic zone (AT hereafter) including the cities of Dakar, Nouakchott, Monronvia and Conakry [Fig1];
– Coastal Guinean zone (GU hereafter) including the cities of Yamoussoukro, Abidjan, Lomé, Abuja, Lagos, Accra, Cotonou and Douala [Fig1]."

To:
" The choice of these regions is coherent with Moron et al. (2016) who used a hierarchical clustering approach to define some blocs of cities over West Africa. The fifteen cities investigated here have been classified in three regions as follows:
– Continental zone (CONT hereafter) including the cities of Bamako, Ouagadougou and Niamey [Fig1];
– Coastal atlantic zone (ATL hereafter) including the cities of Dakar, Nouakchott, Monrovia and Conakry [Fig1];
– Coastal Guinean zone (GU hereafter) including the cities of Yamoussoukro, Abidjan, Lomé, Abuja, Lagos, Accra,
Cotonou and Douala [Fig1].

The CONT and GU regions are very similar to the clusters found by Moron et al. (2016) (see figure below under the title 'Clusters membership'). The ATL region is a specific case because all the cities belonging to the region are not present in the clusters defined by Moron et al. (2016). Therefore, we have investigated the spatial variability of heatwave characteristics for each city in the ATL region. As result, we found coherent evolution between the cities (see [FigS1] in supplement material for

maximum values of T2m using the 90th percentile as threshold); and we put them together to form the ATL bloc."

[Figure]

[Figure]

Besides, the authors note that their interest is in the coastal zone of West Africa region (lines 109). In that case, I am not sure why the analysis of a 'Continental zone' is needed. I would suggest to either rephrase the 'focus' on the coastal area or take out the continental zone analysis.

We rephrased the sentence according to the reviewer's suggestion.

We changed :

"In this study, we are interested in the coastal zone of West Africa, therefore, we identified three regions based on their location and their climate variability on which we conducted our analyses."

to

"In this study, we are interested in the coastal and continental parts of West Africa, therefore, we identified three regions based on their location and their climate variability on which we conducted our analyses."

In section two: Region of interest, Data and Methods, I would suggest the authors rearranging the contents to have only three subsections: 2.1 Region of interest; 2.2 Data and 2.3 Methods (with the corresponding sub-subsections). In 2.2 Data, for instance, I would suggest including the general information on the different reanalysis used (resolution, time-period, climate variables, etc.) as well as the information on the local station data (location, source, time-period, climate variables, percentage of missing values, quality of the series, etc.). Now it is not easy nor clear to find which local information has been used and its characteristics.

We reorganized section 2 according to the reviewer's remarks.

We changed :

"2. Region of interest, Data and Methods
 2.1 Region of interest
 2.2 Heat wave monitoring: Data and indicators
 2.3 Methods "

 to:

"2. Region of interest, Data and Methods
 2.1 Region of interest
 2.2 Data
 2.3 Methods "

We added some clarifications in section 2.2 on the data.

We changed :

[revised manuscript text omitted]

The land sea mask dataset used in this work has been derived from ERA5 reanalysis; it can be accessed on the Copernicus Data Store (CDS). T2m daily maximum and minimum observations at Dakar-Yoff station in Senegal and Aéroport Félix Houphouët Boigny (FHB) station in Ivory Coast have been used to evaluate our interpolation method. This is because we do not have access to other station datasets in the regions. The data from Dakar-Yoff extend from 1 January 1973 to 31 December 2018 containing almost 16% of missing values; and the data from Aéroport FHB are from 1 January 2005 to 31 December 2017 with 0.35% missing values. These data have been provided by some colleagues at Agence Nationale de l'Aviation Civile et de la Météorologie (ANACIM) for the Dakar-Yoff station, and Institut des Géosciences de l'Environnement (IGE) for the Aéroport FHB station."

Although the authors have clearly stated which are the uncertainties that have been studied in the paper, there is a general lack of justification why each number of choices is enough to characterize each uncertainty. Why using only ERA5 and MERRA, for example? There are other reanalyses. Perhaps is it ok to stay with these two, though. In any case, there is a need to better justify whether two are enough to

'characterize' (which implies some sort of specific quantification from a statistical point of view) or, conversely, if they can only be used to 'illustrate' the magnitude of the uncertainties can be important enough to affect the conclusions.

Thanks for this valuable comment.
First of all, we agree with the reviewer that there are others reanalysis products than ERA5 and MERRA; our choice on ERA5 and MERRA-2 to carry this study is supported by some previous work which show that these two reanalyses are part of the most relevant reanalyses used in Africa regions ( e.g. Engdaw et al 2022, Ngoungue et al 2021, Barbier et al 2018). We added this information to the main document.

We changed :
"We use hourly data covering the period going from 1 January 1993 to 31 December 2020 for all the reanalyses."

to:
"We use hourly data covering the period going from 1 January 1993 to 31 December 2020 both for ERA5 and MERRA. Our choice on ERA5 and MERRA to carry this study is supported by some previous work which show that these two reanalyses are part of the most relevant reanalyses used in Africa regions (e.g. , Barbier et al., 2018; Ngoungue Langue et al., 2021; Engdaw et al., 2022)."

The same applies to the uncertainties linked to thresholds and the different ways to define a heatwave. There is a need to better justify and discuss why the authors think their choice of methods and thresholds is enough to map the uncertainties and to which degree this could be achieved.

The choice of the thresholds used to evaluate the uncertainties on heat wave detection is based on previous work. We will justify and discuss our choice of threshold values to conduct the sensitivity analysis in the main document.

We changed :
" As we have seen previously in the section "heat wave detection", the threshold value used for heat waves monitoring has a significant impact on the characteristics of the heat waves. The threshold value is related to the application we want to achieve. In this part of the work, we investigate the sensitivity of heat waves occurrence on different thresholds. To achieve this goal, we define 4 relative

threshold values computed over the entire period: the 75 th , 80 th , 85 th and 90 th daily percentiles "

to :

" As we have seen previously in the section "heat wave detection", the threshold value used for heat waves monitoring has a significant impact on the characteristics of the heat waves. The threshold value is generally tailored to the application one wants to achieve. In this part of the work, we investigate the sensitivity of heat waves occurrence on different thresholds. To achieve this, we define 4 relative threshold values computed over the entire period: i.e. the 75th, 80th, 85th and 90th daily percentiles. The choice of these thresholds to evaluate the changes on heat wave detection is based on previous work. Many studies are using the 90th percentile to define a heat wave (e.g., Fischer and Schär, 2010; Perkins et al., 2012a, b; Déqué et al., 2017; Lavaysse et al., 2018; Barbier et al., 2018); other studies are using the 75th percentile (Guigma et al., 2020). Based on these studies, we decided to test the sensitivity of threshold values from the third quartile (75th) to 90th percentile by steps of 5% to quantify significant changes in the occurrence of heat wave events. As we are studying extreme events, it is not relevant to go below the third quartile; knowing also that this study focuses on human impacts of heat waves, the 90th percentile is enough as a maximum threshold.``

The choice of the methods used to evaluate the sensitivity on heat wave detection is based on previous work. We justified and discussed our choice of the methods to conduct the sensitivity analysis in the main document.

We changed :
"Heat waves are usually defined as consecutive days of extremely hot temperatures above a threshold value of temperature (e.g., Tan et al., 2010; Gasparrini and Armstrong, 2011; Perkins and Alexander, 2013; Wang et al., 2019). Many factors can affect the definition of a heat wave, including the end-user sectors (human health, infrastructures, transport, agriculture) and also the climatic conditions of the regions (Perkins and Alexander, 2013). Therefore, there is no universal and standard definition of a heat wave (Perkins, 2015; Oueslati et al., 2017; Shafiei Shiva et al., 2019). Different thresholds, duration and indicators contribute to divergence in defining heat waves (Smith et al., 2013). Heat waves can be defined from daily meteorological variables such as daily raw temperature ($T_{min}$, $T_{mean}$ and $T_{max}$) (e.g., Fontaine et al., 2013; Beniston et al., 2017; Ceccherini et al., 2017; Déqué et al., 2017; Batté et al., 2018; Barbier et al., 2018; Lavaysse et al., 2018; Engdaw et al.,

2022), mean daily wet bulb temperature (Yu et al., 2021) or heat stress indices (e.g., Robinson, 2001; Fischer and Schär, 2010; Perkins et al., 2012; Guigma et al., 2020) using relative or absolute thresholds. Some other authors use the daily anomalies of temperature to define heat waves (e.g., Stefanon et al., 2012; Barbier et al., 2018). In our case, we use the daily min and max values of: T2m ($T_{2m\,min}$, $T_{2m\,max}$), $T_w$ ($T_{w\,min}$, $T_{w\,max}$), AT ($AT_{min}$, $AT_{max}$) and UTCI ($UTCI_{min}$, $UTCI_{max}$) as indicators for the detection of heat wave events. Three types of heat wave were detected (namely those occurring during daytime, nighttime and both daytime and nighttime concomitantly) using the following methods (see [Fig2]):"

To:

"Heat waves are usually defined as consecutive days of extremely hot temperatures above a threshold value of temperature (e.g.,Tan et al., 2010; Gasparrini and Armstrong, 2011; Perkins and Alexander, 2013; Wang et al., 2019). Many factors can affect the definition of a heat wave, including the end-user sectors (human health, infrastructures, transport, agriculture) and also the climatic conditions of the regions (Perkins and Alexander, 2013). Therefore, there is no universal and standard definition of a heat wave (Perkins, 2015; Oueslati et al., 2017; Shafiei Shiva et al., 2019). Different thresholds, duration and indicators contribute to divergence in defining heat waves (Smith et al., 2013). Heat waves can be defined from daily meteorological variables such as daily raw temperature (Tmin, Tmean and Tmax) (e.g., Fontaine et al., 2013; Beniston et al., 2017; Ceccherini et al., 2017; Déqué et al., 2017; Batté et al., 2018; Barbier et al., 2018; Lavaysse et al., 2018; Engdaw et al., 2022), mean daily wet bulb temperature (Yu et al., 2021) or heat stress indices (e.g., Robinson, 2001; Fischer and Schär, 2010; Perkins et al., 2012a; Guigma et al., 2020) using relative or absolute thresholds. The use of absolute thresholds is very interesting to detect heat waves during the year in regions where the seasonal cycle is well marked. In mid-latitudes for example, the difference of T2m between the summer and winter is very important, approximately +30°C. Using this approach in tropical regions is not suitable, because the seasonal cycle is not so well marked; therefore a relative threshold for heat wave detection is more adapted. Some authors use the daily anomalies of temperature to define heat waves (e.g. Stefanon et al., 2012; Barbier et al., 2018). Most of the previous studies are focused on daytime or nighttime heat waves, ignoring events which occur during the day and night concomitantly. These type of heat waves are very dangerous for human health because the body suffers from heat stress during the day and night (Lavaysse et al., 2018). In our case, we defined 3 methods to detect specific type of heat waves

(namely those occurring during daytime, nighttime and both daytime and nighttime concomitantly) using the daily min and max values of: T2m (T2min, T2max), Tw (Twmin, Twmax), AT (ATmin, ATmax) and UTCI (UTCImin, UTCImax) as indicators. The selected atmospheric variables have been used for heat wave detection in previous studies; they take in account some key variables (air temperature, wind, humidity, radiant temperature) to assess the body heat stress and they are easy to compute. The methods applied are defined below : " ).

There is a need to further explain the downscaling method applied as well as the need for it. The method is not clear, nor how is it applied, as well as which stations were used and why. If I'm not mistaken, the method is applied because the reanalysis products are not enough to go to the city level, and there are not enough stations in the cities of interest to just use point station data. If this is the case, this has to be better explained in the Methods subsection (and, possibly, in the introduction and conclusions sections, too). Hence, I would suggest the authors to expand this section or provide a more detailed description of the methodology as Supplementary material.

We explained in more details the interpolation technique used in this study, which we completed as well in the paper.

We changed :
"Climate models used for weather studies are generally run at global scale, therefore information at local scale is missing in many regions; this is a critical issue. To overcome this problem, downscaling methods can be used. In this work, we studied phenomena at the scale of the city while our products have much coarser spatial resolution. In this context, we need a downscaling approach to attribute variables of interest from global to local scale. Another problem we faced is that most of the cities are located along the coast and influenced by the ocean flow (see [Fig1]). The evaluation of the spatial variability of the correlation between the local scale variable (station) and reanalyses (ERA5), showed high correlation values over the continent [FigS8]. To estimate the proportion of land on a grid point, we used the land sea mask whose values range from 0 to 1. A land sea mask (lsm) of 0 means no land (a point located in the ocean), and a lsm of 1 means that the model cell is fully covered by land. Hence, to estimate the temperature over the city using reanalyses, we use the nearest grid point of reanalyses to the station which satisfies a lsm equal or greater than 0.5 (see [Table3] for lsm values of all the cities considered in this study)."

To:

" Reanalysis dataset used for weather studies are generally run at global scale, therefore information at local scale is missing in many regions; this is a critical issue in regions where there is a lack of observation stations as is the case of African cities. To overcome this problem, sometimes downscaling methods can be used. In this work, we study phenomena at the scale of the cities and reanalyses (ERA5 and MERRA) have too coarse a spatial resolution. The scales of the reanalyses are more representative of the spatial variability of a heat wave occuring in a city than an isolated local stations. Nevertheless, a certain validation must be conducted of testing stations, especially to find the best interpolation technique to estimate local temperature from the reanalyse. This is especially important over the coastal regions. Indeed, most of the cities used in this study are located along the coast and influenced by the ocean air masses (see [Fig1]). The evaluation of the spatial variability of the correlation between the local scale variable (station) and reanalyses (ERA5) for T2m for example, showed high correlation values over the continent [FigS2] (Dakar, Abidjan). This suggests that the station data are well correlated with ERA5 grid points which are located on the continent; so there is a need to know whether a ERA5 grid point is over the continent or not before applying an interpolation technique. To estimate the proportion of land on a grid point, we used the land sea mask (lsm) whose values range from 0 to 1. The land sea mask is a measure of the land occupation on a grid point. A lsm of 0 means no land (a grid point located in the ocean), and a lsm of 1 means that the model cell is fully covered by land. Therefore, to estimate the climate variables over the cities from reanalyses, we use the nearest grid point of reanalyses to the station which satisfies a lsm equal or greater than 0.5 (see [Table1] for lsm values of all the cities considered in this study). This approach was chosen after evaluating different methods such as (see [FigS3 a)] for more details) :

– a bilinear interpolation using the four nearest grid points of reanalyses around the station [FigS3 (a,d)];

– a linear gradient approach which considers that the gradient of temperature is constant between two grid points based on a linear interpolation with a condition on the lsm value (>0.5) [FigS3a (c,f)];

– the selection of the nearest grid point of reanalyses from the station with different values of lsm (>=0.5, 0.75 and 1; we only show for lsm>=0.5) [FigS3a (b,e)] "

- a dynamical downscaling approach taking into account the effect of winds (not shown).

The use of relative thresholds to establish heatwave duration for all the year, though it is systematic, implies that for some regions and periods, the 'heat waves' have different impacts. It could happen that for some regions and periods, though formally there could be a heat wave, in practice, there would not be any impact at all from it. This needs to be highlighted and discussed, to justify that, in any case, the analysis for all the year it is still useful.

We agree with the reviewer point of view about the definition of relative thresholds to process heat waves detection over the year; this is especially true for mid-latitude regions. First of all, our region of interest is West Africa, and in this region the seasonal cycle is not as well marked as in the mid-latitudes. The seasonal thermal amplitude is about 6 °C. Secondly, this study is part of the STEWARd (STatistical Early WArning systems of weather-related Risks from probabilistic forecasts, over cities in West Africa) project which focuses on climate extremes human impacts. Therefore, an estimation of the intensity of heat waves is added by using a fixed yearly threshold (the minimum of the daily climatology percentiles over the period) from which we computed the daily exceedance of hot days. Using this approach, we can clearly evaluate the severity of a heat wave and its potential human impacts which will be higher when occurring during the hottest period of the year . This has been clarified in the manuscript as follows.

We changed :
"Heat waves can be defined from daily meteorological variables such as daily raw temperature (Tmin , Tmean and Tmax ) (e.g., Fontaine et al., 2013; Beniston et al., 2017; Ceccherini et al., 2017; Déqué et al., 2017; Batté et al., 2018; Barbier et al., 2018; Lavaysse et al., 2018; Engdaw et al., 2022), mean daily wet bulb temperature (Yu et al., 2021) or heat stress indices (e.g., Robinson, 2001; Fischer and Schär, 2010; Perkins et al., 2012; Guigma et al., 2020) using relative or absolute thresholds."

To:
" Heat waves can be defined from daily meteorological variables such as daily raw temperature (Tmin , Tmean and Tmax ) (e.g., Fontaine et al., 2013; Beniston et al., 2017; Ceccherini et al., 2017; Déqué et al., 2017; Batté et al., 2018; Barbier et al., 2018; Lavaysse et al., 2018; Engdaw et al., 2022), mean daily wet bulb temperature

(Yu et al., 2021) or heat stress indices (e.g., Robinson, 2001; Fischer and Schär, 2010; Perkins et al., 2012; Guigma et al., 2020) using relative or absolute thresholds. The use of absolute thresholds is well suited to detect heat waves during the year in regions where the seasonal cycle is well marked. In mid-latitudes for example, the seasonal thermal amplitude of T2m is large, approximately 20°C. In tropical regions this method is not suitable since the seasonal thermal amplitude is strongly reduced (6°C). Therefore a relative threshold for heat waves detection is adopted in our study as our region of interest is West Africa."

We change :

" The mean intensity of a heat wave has been defined as the sum of the daily exceedance of daily values of indicators over the daily threshold in a sequence of hot days divided by the total number of heat waves. In the scope of this study, we are interested in human impacts of heat waves, therefore we defined a constant threshold value over the whole period to compute the intensity. "

To :

"The mean intensity of a heat wave has been defined as the sum of the daily exceedance of daily values of indicators to the climatological daily threshold in a sequence of hot days divided by the total number of heat waves. This study is part of the project Agence National de la Recherche STEWARd (STatistical Early WArning systems of weather-related Risks from probabilistic forecasts, over cities in West Africa) project which focuses on climate extremes human impacts. Therefore, the climatological daily threshold is chosen to be constant over the whole period; and it is defined as the minimum of the daily climatology thresholds over the study period. From this approach, we can properly evaluate the severity of a heat wave and its potential impacts on humans."

When performing the comparison through statistical metrics, besides clearly stating that the data is downscaled, I would also suggest comparing the 'downscaled values' with the station ones (whenever possible).

This is done in the validation process of our method through the anomaly of correlation between the different interpolation methods and the station datasets. We added this result to supplement material (see [FigS3a]).

It would also be needed to specify why choosing ERA5 as the reference (instead of MERRA or any other station network).

Thanks to the reviewer for this remark.

The choice of ERA5 as reference for this analysis is based on previous work. For instance, Olauson, 2018 and Ramon et al., 2019 found that ERA5 provides a good representation of various near surface meteorological variables including near surface humidity and wind speed in comparison to others reanalyses including MERRA. However in order to clarify this point, we added in the manuscript some analyses on heatwaves evolution with MERRA reanalysis. Reanalyses are more representative of the spatial variability of the city than a local station.

In the maps at the end, I would suggest using discrete colour bars (continuous ones are not suitable for assigning values). I would also include the cities of interest in all the maps (since this is the focus of the paper).

Thanks to the reviewer for this valuable suggestion. We plot the maps accordingly.

We changed:

[Figure]

To:

[Figure]

We changed :

[Figure]

**Figure 5.** *Evolution of the heat wave duration with respect to the threshold values using T2m as indicator respectively for : a-c) ERA5 and d-f) MERRA. The slope is computed using the $75^{th}, 80^{th}, 85^{th}$ and $90^{th}$ percentiles. X and Y-axis respectively represent the longitude and latitude in degrees. The color bar shows the values of the linear evolution of the duration of heat wave per percentile. The white blanks indicate non significant changes in the duration of heat waves per percentile.*

To :

[Figure]

**Figure 5**. *Evolution of the heat wave duration with respect to the threshold values using T2m as indicator respectively for : a-c) ERA5 and d-f) MERRA. The slope of the regression line in day per percentile is computed by fitting a linear regression between the threshold values (75, 80, 85, 90) and their corresponding heat waves's duration (D75, D80, D85, D90) . X- and Y- axis respectively represent the longitude and latitude in degrees. The color bar shows the values of the slope. The white blanks indicate non-significant changes in the duration of heat waves per percentile.The significance of the slope of the regression line has been computed using a two-sided Chi-square test.*

Figure 4 depicts the slope of the linear regression in heatwaves (also figure 5). The caption says that the slope is computed using the 75th, 80th, 85th and 90th percentiles, but there is only one map per variable and reanalysis. Does this mean that the trend is computed for all the four percentiles simultaneously? This is not very clear when reading the methods section, and a rephrasing and/or extension of the description would be advisable. Besides, there is also a need to state more clearly (both in the methods and in the captions) which method has been applied to compute the significance of the slope. That said, if the slope is computed with all the four thresholds simultaneously, it wouldn't be a conventional approach and, consequently, a more thorough justification about its correctness and its utility would be needed (compared to performing the analysis independently for each threshold).

Thanks to the reviewer for this comment.

The assessment of changes in heat waves occurrence or duration with respect to the threshold (75th, 80th, 85th, 90th percentiles) is processed independently for the 4 thresholds; this is done by the computation of the linear evolution coefficient over each grid point. The linear evolution coefficient is defined as the slope of the linear regression line fitted between the threshold values (Q75, Q80, Q85, Q90) and the number of events associated to each threshold (N75, N80, N85, N90) or their

corresponding duration (D75, D80, D85, D90). In fact, for each grid point, a regression line is firstly fitted between the threshold values and their corresponding occurrence or duration; and secondly the slope is computed. We make it clear in the manuscript.

We changed:

" To quantify the changes of heat waves occurrence with respect to the threshold values, we analyse for each grid point the linear evolution of events detected and their duration. The linear evolution is computed by fitting a linear regression between the threshold values (75, 80, 85, 90) and the number of events associated to each threshold (N1, N2, N3, N4) or their corresponding duration (D1, D2, D3, D4). We are aware that this regression based on 4 points is not very robust, nevertheless it makes it possible to obtain information on the evolution of the heat wave characteristics with respect to the thresholds. Therefore, we evaluated the significance of the slope values according to the thresholds using a confidence level of 95%. "

To:

" The sensitivity in heat waves occurrence or duration with respect to the threshold (75th, 80th, 85th, 90th percentiles) is processed simultaneously for the 4 thresholds; this is done by the computation of the linear evolution coefficient over each grid point. The linear evolution coefficient is defined as the slope of the linear regression line fitted between the threshold values (Q75, Q80, Q85, Q90) and the number of events associated to each threshold (NQ75, NQ80, NQ85, NQ90) or their corresponding duration (DQ75, DQ80, DQ85, DQ90). The computation of the linear evolution coefficient is done by the following steps:

- After processing to heat waves detection at each grid point for the 4 thresholds separately, we compute for each of them the heat waves frequency and duration;

- then fitted a regression line between the threshold values (Q75, Q80, Q85, Q90) and their corresponding occurrence or duration. This is done for each grid point;

- Finally, the changes in heat waves occurrence/duration from the 75th to 90th percentiles at each grid point, is evaluated by the computation of the slope of the regression line fitted at step 2 between the threshold values and their corresponding heat waves occurrence/duration.

We are aware that this regression based on 4 points is not very robust, nevertheless it makes it possible to obtain information on the evolution of the heat wave characteristics with respect to the thresholds. Therefore, we evaluated the significance of the slope values according to the thresholds using a confidence level of 95%. The significance of the slope has been evaluated using a two sided Chi-square statistics test (Pandis, 2016)."

**Technical corrections**

Figure 6 lacks titles in the top row
This is correct, we add titles in the top row.

Figures should include units when necessary. For example, in figure 1, 'meters above sea level; in figure 6 it would be 'number of days' or 'Number of events / occurrences'); in figure 3 and figure 4, number of events or days / year; figure 7; figure 11...
We add units in figures when necessary.

It is not clear how figure 2 has been obtained. Is it built with data from all the stations?Cities? Grid-points? It is just an illustration for a single grid-point? This has to be included in the caption (as well as in the main text).
In fact, figure2 represents a schematic illustration of the different types of heat waves analysed in this paper. It is not obtained from a specific station nor grid point.

We changed:
"Figure 2. Detection process of heat wave: HW1/HW2 represent events associated respectively to maxima/minima temperature, HW3 are events detected at same time in maxima and minima temperatures. The red/blue lines with circles are max/min daily temperatures. Red/blue solid lines are respectively max/min thresholds. X- and Y- axis represent the time in days and the temperature in degrees celsius. 'With pool' refers to the pooling of two (or more) events separated by a day characterized by the value of a given indicator below the daily XX th percentile."

To :

"Figure 2. Detection process of heat wave: HW1/HW2 represent events associated respectively to maxima/minima temperature, HW3 are events detected at the same time in maxima and minima temperatures. The red/blue lines with circles are max/min daily temperatures. Red/blue solid lines are respectively max/min thresholds. X- and Y- axis represent the time in days and the temperature in degrees celsius. 'With pool' refers to the pooling of two (or more) events separated by a day characterized by the value of a given indicator below the daily XXth percentile. This figure is a 'schematic' illustration of the different types of heat waves investigated in this work"

Sometimes x- axis and y- axis is written is capital letters and sometimes it is not.
The acronyms for variables should be the same in figures (titles, for instance), captions as well as in the main text.
We corrected the manuscript accordingly. We changed  x- axis and y- axis  to capital letters everywhere in the document.

Column titles in figure 8 and 9 are difficult to understand. Besides, the idea to display different parameters in the same format it is confusing (apparently, from the caption, 2nd, 3rd and 4th columns display percentages instead of duration of heatwaves). I would suggest to only maintain the same format when displaying the same elements.

Thanks to the reviewer for this suggestion, we reorganized the figure 8 and 9 accordingly.

We changed:

[Figure]

**Figure 8.** *Seasonal variability of heat wave yearly duration using maximum values of indicators: a) T2m, b) TW and c) AT. The first column shows the evolution of heat wave duration per year over the whole period 1993-2020. The $2^{nd}(d,e,f)$, $3^{rd}(g,h,i)$ and $4^{th}(j,k,l)$ columns represent respectively the contribution in percentage of the sub-periods 1993-2001, 2002-2011 and 2012-2020 to heat wave duration over the whole period. We compute a a 3-month running mean to smooth the seasonal cycle. The X-axis represents the time in days; and the Y-axis respectively the duration of heat waves, and the contribution of each decade. Red/blue/green lines represent the evolution of heat wave duration over CONT/AT/GU regions (see region of interest section for more details).*

To :

[Figure]

[Figure]

Figure 8. Seasonal variability of heat waves characteristics using maximum values of T 2m, T w, AT : a-c) duration and d-f) intensity. We compute a 3-month running mean to smooth the seasonal cycle. The detection of heatwaves is done using the 90th percentile as threshold over : CONT (a − d), ATL (b − e), GU (c − f ) regions. Red/blue/green strong and dashed lines represent respectively the results using T2m, Tw, AT from ERA5 and MERRA. The Y- and X- axis represent the duration and intensity of heat waves and the time in month respectively.

We did the same for Figure 9 (not shown here)

In figure 10 it is not clear what those percentages refer to. Are percentages from the total of days? From the total of heat wave days? Do they have to sum 1 in total? The

phrase 'using maximum values of indicators based on the duration' is not very clear, either. What does this refer to? The thresholds? The methods? The variables? This also extends to the other figures applying the same approach

Thanks to the reviewer for this comment; we clarify all these points in the paper.

Figure10 represents the classification in terms of duration of heat waves detected with the 90 th percentile as threshold using maximum values of indicators (T2m,Tw and AT) over the period 1993-2020. Firstly, we detect heat waves and compute their duration; after we construct clusters of heat waves based on their duration (3d, 4d-6d, 7d-9d, 10d-12d, +13d) and finally, we quantify the proportion of each class of heat waves to the total number of events detected.

We changed :
"Figure 10. Classification of the heat waves detected using maximum values of indicators based on the duration: a) T 2m, b) T W and c) AT. The X and Y-axis represent respectively the percentage of the heat wave per class and the duration in day. Red/blue/green bars represent the percentage of heat waves detected over CONT/AT/GU regions (see region of interest section for more details)."

To:
"Figure 10. Classification of the heat waves detected with the 90th percentile as threshold using maximum values of indicators based on their persistence over the period 1993-2020 : a) T2m, b) Tw and c) AT. Firstly, we detect heat waves and compute their duration; after we construct clusters of heat waves based on their duration (3d, 4d-6d, 7d-9d, 10d-12d, +13d) and finally, we quantify the proportion of each class of heat waves to the total number of events detected. The Y- and X- axis represent respectively the percentage of the heat waves per class and the duration in day. Red/blue/green bars represent the percentage of heat waves detected over CONT/ATL/GU regions (see region of interest section for more details). The sum of the contribution of heat waves in different clusters is equal to 1 for each region."

---

## Author Comment (AC2)

**Nat. Hazards Earth Syst. Sci. Discuss., referee comment RC2**
**https://doi.org/10.5194/nhess-2022-192-RC2, 2022**
**Comment on nhess-2022-192**
Anonymous Referee #2

Referee comment on "Heat waves monitoring over West African cities: uncertainties, characterization and recent trends" by Cedric Gacial Ngoungue Langue et al., Nat. Hazards
Review – Heat waves monitoring over West African cities : uncertainties, characterization and recent trends by Ngoungue Langue et al.

This manuscript discusses the uncertainties related to the use of reanalysis datasets for the monitoring and recent evolution of heat wave indices over West Africa, with a focus on several urban areas.

Three types of uncertainties are addressed, that stemming from the dataset itself, the choice of the threshold for heat wave definition, and the type of indicator (minimum or maximum temperature, including or not the influence of other meteorological variables such as surface winds or humidity).

The results presented are consistent with several past works on the topic, adding recent years and in some aspects providing different diagnostics than the analyses previously published (e.g. Cecchirini et al. (2017), Moron et al. (2015), Barbier et al. (2018) - to cite only a few focused on West Africa). However, the manuscript in its present form has a number of shortcomings and fails to present results in a clear and concise way. I found some parts of the manuscript difficult to follow, and the number of figures (including all those in the supplemental) leave me with the impression that extracting key conclusions from the numerous statistics computed was a challenging, but uncompleted, task for the authors. Also missing from the manuscript, with respect to the title of the submission, is a clear description of what the authors are aiming for in using reanalysis data for city or district-level monitoring. The conclusions in terms of reanalysis uncertainties clearly point major caveats to such an approach, which then weakens the key messages of the paper.

Thank you to the reviewer for these pertinent comments. We have taken into consideration many of the recommendations proposed by the reviewer, and added

some analyses. Some points such as the interpolation method applied in the study, the classification of cities into different climate regions and the use of reanalysis data instead of local station data have been clarified in the manuscript. We are confident that all these changes to the paper will improve its quality.

That said, I am confident that the authors can revise their submission, taking into account major comments listed below, and turn this manuscript into a valuable contribution.

**Major comments**

The manuscript dwells quite some time on the description of differences between the MERRA and ERA5 reanalyses, evidencing large uncertainties between both. Then, ERA5 is kept as "truth" for the rest of the paper (from section 3.3). Differences between heat indices over West Africa computed from reanalyses and other sources of data and related uncertainties have been discussed in a number of past publications (Barbier et al. 2018, Batté et al. 2018, Engdaw et al. 2022...). Why is ERA5 thought to be better a reference than MERRA, and why is reanalysis kept as a suitable source of data for the rest of the analysis?

Thanks to the reviewer for this remark.
The choice of ERA5 as reference for this analysis is based on previous work. For instance, Olauson, 2018 and Ramon et al., 2019 found that ERA5 provides a good representation of various near surface meteorological variables including near surface humidity and wind speed in comparison to others reanalyses including MERRA. However in order to clarify this point, we added in the manuscript some analyses on heatwaves evolution with MERRA reanalysis. Reanalyses are more representative of the spatial variability of the city than a local station.

Did you find similar caveats in the MERRA dataset than those highlighted by Engdaw et al. 2022?
Engdaw et al. 2022 found that MERRA in comparison to the other datasets (NCEP, ERA5, GS0D..), overestimates the evolution of heatwaves indices during the 2000s. In our analyses, we did not find this specific overestimation of heatwave characteristics with MERRA. However some discrepancies are noticed between MERRA and ERA5.

You (very briefly) mention station data in Supplemental Fig S8, but did you perform some assessment of ERA5 versus MERRA with respect to station data corresponding to your cities of interest? How adequate is ERA5 to represent heatwaves in these cities? All of this is left to the reader to guess or infer, which is a bit confusing given the title of the manuscript.

Thanks to the reviewer for this remark.
We conducted some analyses on ERA5 versus MERRA with respect to station data with the nearest grid to the station with a land sea mask greater or equal to 0.5. We found that ERA5 shows slightly better correlation to the station data than MERRA (see the [FigS3b] below).

[Figure]

I'm a bit puzzled by the mention of station data and the separation of the region of interest in climatic regions, and the use of gridded reanalysis data in the study. It wasn't fully clear to me upon reading the manuscript whether the approach used was completely validated. In the supplement, there is a figure (S8) which appears to tackle this question, but it is only very briefly mentioned in the manuscript. The authors furthermore say they find high levels of correlation, but I would argue this is only the case for Tmax over Dakar out of the four results shown.

We clarified this point in the response of the comment related to section 2.3.1

The classification of cities should be described in more detail, and justified. Indeed the classes found are used to compute composites of characteristics in section 3.4, but this approach will be valid only if there is indeed some level of consistency between the cities. Given the spatial distribution of cities of interest, some will likely be characterized by neighboring grid points from the reanalysis, whereas others are much more distant. I'm missing a clear justification of why this approach is valid.

We clarified this point in the manuscript.
We changed:
"The choice of these regions has been validated by conducting some analyses over the cities belonging to each region (not shown). The repartition of the different climatic regions is given as follows :
– Continental zone (CONT hereafter) including the cities of Bamako, Ouagadougou and Niamey [Fig1];
– Coastal atlantic zone (AT hereafter) including the cities of Dakar, Nouakchott, Monronvia and Conakry [Fig1];
– Coastal Guinean zone (GU hereafter) including the cities of Yamoussoukro, Abidjan, Lomé, Abuja, Lagos, Accra, Cotonou and Douala [Fig1]."

To:
" The choice of these regions is coherent with Moron et al. (2016) who used a hierarchical clustering approach to define some blocs of cities over West Africa. The fifteen cities investigated here have been classified in three regions as follows:
– Continental zone (CONT hereafter) including the cities of Bamako, Ouagadougou and Niamey [Fig1];
– Coastal atlantic zone (ATL hereafter) including the cities of Dakar, Nouakchott, Monrovia and Conakry [Fig1];
– Coastal Guinean zone (GU hereafter) including the cities of Yamoussoukro, Abidjan, Lomé, Abuja, Lagos, Accra,
Cotonou and Douala [Fig1].

The CONT and GU regions are very similar to the clusters found by Moron et al. (2016) (see figure below under the title 'Clusters membership'). The ATL region is a specific case because all the cities belonging to the region are not present in the clusters defined by Moron et al. (2016). Therefore, we have investigated the spatial variability of heatwave characteristics for each city in the ATL region. As result, we found

coherent evolution between the cities (see [FigS1] in supplement material for maximum values of T2m using the 90th percentile as threshold); and we put them together to form the ATL bloc."

[Figure]

[Figure]

A final point more related to editing is the numbering and order of figures. The figures (including supplemental figures) should be numbered according to the order of appearance in the text. If not, the reader has to go back and forth between figures and this makes the paper tedious to read.

We reorganized the numbering and order of the figures as they appear in the manuscript.

**Specific comments**

**Abstract**

What is the main goal of the manuscript? Already from reading the (quite long) abstract, it appears that the scientific questions are not very specific.
We rearranged the abstract as suggested in the comment.

We changed :
"Heat waves can be one of the most dangerous climatic hazards affecting the planet; having dramatic impacts on the health of humans and natural ecosystems as well as on anthropogenic activities, infrastructures and economy. Based on climatic conditions in West Africa, the urban centers of the region appear to be vulnerable to heat waves. In this study, we assess the potential uncertainties encountered in the process of heat waves monitoring and analyse their recent trend in West Africa cities. This is investigated using two state-of-the-art reanalysis products namely ERA5 and MERRA for the period 1993-2020. Three types of uncertainties are discussed. The first type of uncertainty is related to the reanalyses themselves, with MERRA showing a cold bias with respect to ERA5 over the Sahel and Guinean regions except over some countries (Guinea Bissau, Sierra Leone,Liberia). Furthermore, large discrepancies are found in the representation of extreme values in the reanalyses over the southern Sahel and the Guinea coast. The second type of uncertainty is related to the sensitivity of heat waves frequency to the threshold values used to monitor them. Heat waves detected using the lowest threshold value are very persistent and last for several days; while the duration of heat waves related to high threshold values is shorter. The choice of indicators and the methodology used to define heat waves constitutes the third type of uncertainty. Three sorts of heat waves have been analysed, namely those occurring during daytime, nighttime and both daytime and nighttime concomitantly. Four indicators have been used to analyse heat waves based on 2-m temperature, humidity, 10-m wind or a combination of these. Nighttime and daytime heat waves are in the same range of occurrence while concomitant day- and nighttime events are extremely rare

[revised manuscript text omitted]

l. 65: Either explain more how this result is important (if relevant for your work) or shorten the paragraph.

We added some explanations on this point in the manuscript.
We changed:
"Another important result of this work, is the radiative effect of water vapor on minimum temperatures during the heat wave period."
To:
"Another important result of this work is the radiative effect of water vapor on minimum temperatures during the heat wave period.  This can lead to extreme heat conditions during the night and cause death to eldery. "

**Region of interest, data and methods**

l. 110: "The choice of these regions has been validated by conducting some analyses over the cities belonging to each region (not shown)."
This is a shame, since it clearly is a key aspect in your use of this regional scale in the analyses that follow, and links to the title of the manuscript (see one of my major comments above).
This point has been clarified in response to the major comments.

l. 124: The authors restrict their analysis to 1993-2020. Both MERRA and ERA5 data are available before 1993, and statistics would likely be more robust by including more years. Is there a specific reason for this?
We agree with the reviewer that MERRA and ERA5 data are available before 1993, and also that the statistics will be more robust with more years. The idea at the beginning of this study was to do a comparative analysis with the seasonal forecast models (hindcasts) that are commonly available since 1993.  In this study, we do not focus on the climatic evolution of heat waves but on heat wave processes. Therefore, a period of 28 years is already sufficient for the analyses.

Section 2.3.1: As stated earlier, I think this section leaves a lot of crucial points of the study partially hidden to the reader, which weakens the conclusions.
We explained in more detail the approach used for the estimation of atmospheric variables at the local scale  in this study, which we completed as well in the paper.

We changed :
"Climate models used for weather studies are generally run at global scale, therefore information at local scale is missing in many regions; this is a critical issue. To

overcome this problem, downscaling methods can be used. In this work, we studied phenomena at the scale of the city while our products have much coarser spatial resolution. In this context, we need a downscaling approach to attribute variables of interest from global to local scale. Another problem we faced is that most of the cities are located along the coast and influenced by the ocean flow (see [Fig1]). The evaluation of the spatial variability of the correlation between the local scale variable (station) and reanalyses (ERA5), showed high correlation values over the continent [FigS8]. To estimate the proportion of land on a grid point, we used the land sea mask whose values range from 0 to 1. A land sea mask (lsm) of 0 means no land (a point located in the ocean), and a lsm of 1 means that the model cell is fully covered by land. Hence, to estimate the temperature over the city using reanalyses, we use the nearest grid point of reanalyses to the station which satisfies a lsm equal or greater than 0.5 (see [Table3] for lsm values of all the cities considered in this study)."

To:

" Reanalysis dataset used for weather studies are generally run at global scale, therefore information at local scale is missing in many regions; this is a critical issue in regions where there is a lack of observation stations as is the case of African cities. To overcome this problem, sometimes downscaling methods can be used. In this work, we study phenomena at the scale of the cities and reanalyses (ERA5 and MERRA) have too coarse a spatial resolution. The scales of the reanalyses are more representative of the spatial variability of a heat wave occuring in a city than an isolated local stations. Nevertheless, a certain validation must be conducted of testing stations, especially  to find the best  interpolation  technique to estimate local temperature from the reanalyse. This is especially important over the coastal regions. Indeed, most of the cities used in this study are located along the coast and influenced by the ocean air masses (see [Fig1]). The evaluation of the spatial variability of the correlation between the local scale variable (station) and reanalyses (ERA5) for T2m for example, showed high correlation values over the continent [FigS2] (Dakar, Abidjan). This suggests that the station data are well correlated with ERA5 grid points which are located on the continent; so there is a need to know whether a ERA5 grid point is over the continent or not before applying an interpolation technique. To estimate the proportion of land on a grid point, we used the land sea mask (lsm) whose values range from 0 to 1. The land sea mask is a measure of the land occupation on a grid point. A lsm of 0 means no land (a grid point located in the ocean), and a lsm of 1 means that the model cell is fully covered by land. Therefore, to estimate the climate variables over the cities from

reanalyses, we use the nearest grid point of reanalyses to the station which satisfies a lsm equal or greater than 0.5 (see [Table1] for lsm values of all the cities considered in this study). This approach was chosen after evaluating different methods such as (see [FigS3a] for more details) :

– a bilinear interpolation using the four nearest grid points of reanalyses around the station [FigS3a (a,d)];

– a linear gradient approach which considers that the gradient of temperature is constant between two grid points based on a linear interpolation with a condition on the lsm value (>0.5) [FigS3a (c,f)];

– the selection of the nearest grid point of reanalyses from the station with different values of lsm (>=0.5, 0.75 and 1; we only show for lsm>=0.5) [FigS3a (b,e)] "

- a dynamical downscaling approach taking into account the effect of winds (not shown).

l. 157: Did you compare the nearest neighbor strategy with lsm > 0.5 to the station data? Of course station data will be representative of temperature at a very local scale, but on the other hand, resolution of the reanalyses is quite coarse when compared to cities.

Thanks to the reviewer for this question.
The nearest neighbor strategy with lsm > 0.5 has been compared to the station data through a correlation analysis (see [Fig S3.a] in the supplement material). We took it into account in the previous comment.

Section 2.3.3 and heat wave duration computation

I was confused by the equation l. 200 and the explanation.
In lines 196-199 you explain that heat wave duration is computed as the mean over the number of heat waves of the total number of hot days in heat waves (I agree with this definition). But then when describing the equation terms, it appears you count all of the hot days whether belonging to a heat wave or not. If d is the number of hot days, then shouldn't $\delta_j$ in the sum be an indicator of the day belonging to a heat wave rather than the corresponding temperature exceeding the 90th percentile (this condition is already fulfilled for a hot day...)?

Thanks to the reviewer for this comment. When computing the duration of heatwaves, we take into account just the hot days belonging to heatwaves. Yes we agree with the reviewer that the kronecker δj here is an indicator of day belonging to a heatwave not the temperature exceedance from the 90th percentile. 'd ' is the number of hot days in heatwaves. We clarified this point in the manuscript.

We changed :

"N represents the total number of heatwaves per grid point and 'd' the number of hot days."

To :

"N represents the total number of heatwaves per grid point and 'd' the number of hot days in the heatwave events. The kronecker δj is used here because we pooled heatwaves separated by 1 day together to form single events. For example, two heatwaves with respective duration of 4 and 3 days separated by 1 day below the threshold will be counted as 1 event with a duration of 7 days. "

Later in the manuscript, it wasn't clear to me why mean duration could be lower than 3 days (for instance in Fig. 7), since your criteria to define a heat wave is for having at least 3 consecutive days above the given threshold. I may have missed something here, but in any case, this needs clarification in the methods section.

Thanks to the reviewer for this remark.

We clarified this point in the manuscript, and we agree with the reviewer. As the min duration of heat waves is supposed to be 3 days, this implies that the mean duration will be greater or equal to 3 days ( see [Fig S17] in the supplement material).

Section 2.3.4

You define POD but then refer to "hit rate" when discussing the results and in Fig. S1. More generally, in your definitions of the statistical metrics, you use the terms "forecast system" and "observations". Implicitly, later in your discussion of results, ERA5 is often the "observation" and MERRA2 the "forecast system", but I would argue that these terms are quite misleading and suggest you rather use terms like "evaluated dataset" and "reference".

Thanks to the reviewer for this suggestion, we correct this point accordingly.

We changed :

"The POD, also known as the "hit", is a measure of the fraction of events detected by a forecast system knowing that the events happen in the observations at the same time."

To :

"The Hits rate, also known as the "hit", is a measure of the fraction of events detected in an evaluated dataset knowing that the events happen in the reference at the same time. Some previous work such as (Olauson, 2018; Ramon et al., 2019) found that ERA5 provides a good representation of various near surface meteorological variables including near surface humidity and wind speed in comparison to others reanalyses including MERRA. Therefore for the computation of the hit, we choose ERA5 as the reference and MERRA the evaluated dataset"

**Results**

Fig. 3: The blue/red color scale for figures a) and b) isn't the best choice.

Thanks to the reviewer for this remark. We changed the color in Fig 3 a) and b) accordingly.

We changed :

[Figure]

To:

[Figure]

l. 263: It would be worth specifying either here on in section 2.3.4 for what event the scores are computed (hot days).

Thanks to the reviewer for this remark. We added this information to the manuscript. We changed:

" The hit rate and GSS have been computed using T2m "

" The coherence of reanalyses at regional scale has been evaluated using statistical metrics such as the probability of detection (POD), anomaly of correlation (ACC) and the Gilbert skill score (GSS)"

To:

" The hits rate and GSS have been computed in terms of hot days using T2m"

"The coherence of reanalyses at regional scale has been evaluated using statistical metrics such as the hits rate, anomaly of correlation (ACC) and the Gilbert skill score (GSS). The hits rate and GSS are used to evaluate hot days in the reanalyses."

l. 281: "changes of heat waves occurrence": What do you call occurrence? The total number of events over the period of study?

We clarified it in the manuscript.

We changed :

"To quantify the changes of heat waves occurrence with respect to the threshold value"

To:

" To quantify the changes of heat waves frequency with respect to the threshold value"

l. 308: "we use the 90th for heat wave analyses" → you mean the 90th percentile?

Thanks to the reviewer for this remark. We took it into account in the manuscript accordingly.

We changed:

l. 308: "we use the 90th for heat wave analyses"

To:

l. 308: "we use the 90$^{th}$ percentile for heat wave analyses"

l. 321: "Tw takes in account the effect of humidity on the temperature" → I would argue this is also the case for AT, which includes this influence through the term related to water vapor pressure.

We change

"Tw takes in account the effect of humidity on the temperature"

To :

"It appears that Tw is more sensitive to humidity than the other indicators (see formula of Tw). "

l. 332-334: These sentences introduce a new aspect of results, I would therefore recommend moving this to section 3.4. By the way, the numbering of Fig.12 should be Fig. 7.

At this stage of the manuscript, we want to provide a summary of the findings in this section. Therefore, we kept these sentences at the end of this section.

l. 336: "CONT, AT and GU see section "region of interest" for more details" → As a reader I was frustrated at this stage since the details in the section to which you refer doesn't provide these details (it is even stated "not shown").

We already took this remark into account previously (see response to the comment on the classification of cities)

l. 344: "The heat waves detected in the GU region have a short duration and a weak intensity [Fig 7]" → As mentioned earlier, I was surprised that duration is lower than 3 whereas by your definition heatwaves should last a minimum of three days to be

considered as such. Maybe the values are divided by the number of cities? This is a clear blind spot in your methodology. Please clarify this (also in the figure legend).

We clarified this point in the manuscript, and we agree with the reviewer. As the min duration of heat waves is supposed to be 3 days, this implies that the mean duration will be greater or equal to 3 days ( see figure S17 in the supplement material).

l. 364-374: Splitting the (already short) period into yet shorter sub-periods calls for some comment on the robustness of the analysis, especially since other factors may influence the occurrence of heat waves (e.g. El Nino, decadal variability, ...)

Thanks to the reviewer for this comment. The idea behind the splitting of the study period was to quantify the evolution of heatwaves during the last 3 decades in a context of global warming.

**Discussion**

Regarding the differences between ERA5 and MERRA2, Engdaw et al. (2021) identify striking differences between MERRA and other reanalysis and observational datasets in the 2000s for heatwave indices. MERRA appears to be a clear outlier. Did you look into this and draw similar conclusions?

Engdaw et al. 2022 found that MERRA in comparison to the other datasets (NCEP, ERA5, GS0D..), overestimates the evolution of heatwaves indices during the 2000s. In our analyses, we didn't find this specific overestimation of heatwave characteristics with MERRA. However some discrepancies are noticed between MERRA and ERA5.

l. 412: The correspondence between heatwaves and El Nino events was suggested in Moron et al. 2016 which you could include in your introduction and at this stage of the discussion.

Thanks to the reviewer for this suggestion, we included it in the manuscript.

**Typos and editing suggestions**

l. 64: CRNM → CNRM

Modify as suggested

AT is used as an abbreviation both for apparent temperature and the Atlantic cities

Thanks to the reviewer for this remark.

We change:

"AT region"

To:

"ATL region"

l. 280: "see [Fig S3] Fig. S3 in the supplemental material"

Thanks to the reviewer, we took it into account.

We changed:

"see [Fig S3] Fig. S3 in the supplemental material"

To:

"see [Fig S3] in the supplemental material"

Please harmonize the notations used and specify carefully each notation: for example Ws is wind speed in the AT equation, this is never specified. What is Ta in the same equation?

Thanks to the reviewer for the remark. We clarified it in the manuscript.

We changed:

"Where T(°C), Td(°C), p(hPa) and q are respectively the ambient temperature, dew-point temperature, pressure and specific humidity"

To:

"Where T(°C), Td(°C), T0(K), p(hPa), Ws(m/s) and q are respectively the ambient temperature, dew-point temperature, reference temperature, pressure, wind speed and specific humidity."

Table 2, C2: typo peristent → persistent

Thanks to the reviewer, we took it into account in the manuscript.

| Classes | Duration (days) | Degree of persistence |
|---------|-----------------|----------------------|
| C1 | 3 | normal |
| C2 | 4-6 | persistent |
| C3 | 7-9 | very persistent |
| C4 | 10-12 | severe |
| C5 | +13 | very severe |

Figure 6: Top row figure titles are missing

We corrected it in the manuscript.

[Figure]

Figure S16: "incertitude" → you mean uncertainty?

We corrected it in the manuscript.

[Figure]

Overall the manuscript requires careful proofreading (watch out for missing parentheses and brackets).

Thanks to the reviewer for this reamark, we took it into account in the manuscript.

The figure captions should also be revised carefully, and include information on the datasets used (the reader shouldn't have to dig for this information in the text).

Thanks to the reviewer for this advice, we corrected the captions accordingly.

Suggested reference:

Moron, V. et al. (2016) Trends of mean temperatures and warm extremes in northern tropical Africa (1961–2014) from observed and PPCA-reconstructed time series. J. Geophys. Res. Atmos., 121, 5298–5319, doi:10.1002/2015JD024303.

Thanks to the reviewer for this suggestion, we added it to the manuscript.